# Differentially Private Model Personalization

**Prateek Jain**
Google Research
prajain@google.com

**Keith Rush**
Google Research
krush@google.com

**Adam Smith**
Boston University
ads22@bu.edu

**Shuang Song**
Google Research
shuangsong@google.com

**Abhradeep Thakurta**
Google Research
athakurta@google.com

## Abstract

We study personalization of supervised learning with user-level differential privacy. Consider a setting with many users, each of whom has a training data set drawn from their own distribution $P_i$. Assuming some shared structure among the problems $P_i$, can users collectively learn the shared structure—and solve their tasks better than they could individually—while preserving the privacy of their data? We formulate this question using joint, *user-level* differential privacy—that is, we control what is leaked about each user's entire data set.

We provide algorithms that exploit popular non-private approaches in this domain like the Almost-No-Inner-Loop (ANIL) method, and give strong user-level privacy guarantees for our general approach. When the problems $P_i$ are linear regression problems with each user's regression vector lying in a common, unknown low-dimensional subspace, we show that our efficient algorithms satisfy nearly optimal estimation error guarantees. We also establish a general, information-theoretic upper bound via an exponential mechanism-based algorithm.

## 1   Introduction

Modern machine learning techniques are amazingly successful but come with a range of risks to the privacy of the personal data on which they are trained. Complex models often encode exact personal information in surprising ways—allowing, in extreme cases, the exact recovery of training data from black box use of the model [7, 8]. The emerging architecture of modern learning systems, in which models are trained collaboratively by networks of mobile devices using extremely rich, personal information exacerbates these risks.

The paradigm of *model personalization*, a special case of multitask learning, has emerged as one way to address both privacy and scalability issues. The idea is to let users train models on their own data—for example, to recognize friends' and family members' faces in photos, or to suggest text completions that match the user's style—based on information that is common to the many other similar learning problems being solved by other users in the system. Even a fairly limited amount of shared information—a useful feature representation or starting set of parameters for optimization, for example—can dramatically reduce the amount of data each user requires. But that shared information can nevertheless be highly disclosive.

In this paper, we formulate a model for reasoning rigorously about the loss to privacy incurred by sharing information for model personalization. In our model, there are $n$ users, each holding a dataset of $m$ labeled examples. We assume user $j$'s data set $D_j$ is drawn i.i.d. from a distribution $P_j$; the user's goal is to learn a prediction rule that generalizes well to unseen examples from $P_j$. Ideally, the user should succeed much better than they could have on their own. We give new algorithms for this setting, analyze their accuracy on specific data distributions, and test our results empirically.

35th Conference on Neural Information Processing Systems (NeurIPS 2021).

We ask that our algorithms satisfy *user-level, joint differential privacy* (DP) [28] (called *task-level* privacy, in the context of multi-task learning [32]). In this setting, each user provides their data set $D_j$ as input to the algorithm and receives output $A_j = A_j(D_1, ..., D_n)$. We require that for every choice of the other data sets $D_{-j} = (D_1, ..., D_{j-1}, D_{j+1}, ..., D_n)$ and for every two data sets $D_j$ and $D'_j$, the collective view of the other users $A_{-j}$ be distributed essentially identically regardless of whether user $j$ inputs $D_j$ or $D'_j$. The standard model of differential privacy doesn't directly fit our setting, since the model ultimately trained by user $j$ will definitely reveal information about user $j$'s data set. That said, the algorithms we design can ultimately be viewed as an appropriate composition of modules that satisfy the usual notion of DP (an approach known as the *billboard* model). For simplicity, we describe our algorithms in a centralized model in which the data are stored in a single location, and the algorithm $\mathcal{A}$ is run as a single operation. In most cases, we expect $\mathcal{A}$ to be run as a distributed protocol, using either general tools such as multiparty computation or lightweight, specialized ones such as differentially private aggregation to simulate the shared platform.

Intuitively, strong privacy requirement at user level, while still demanding that users share some common information is significantly challenging. For one, as each user individually has a small amount of data, it has to share information about it's model/data to learn a meaningful representation. Furthermore, in practical personalization settings, there is feedback loop between the common or *pooled* knowledge of all users and the personalized models for each user. That is, starting with reasonable personalized models for each user, leads to a better pooled information, while good pooled information then helps each user learn better personal model. Now, requirement of strong privacy guarantees forces the pooled information quality to degrade up to some extent, which can then lead to poorer personalized model and form a negative feedback loop.

## 1.1 Contributions

We consider two types of algorithms for DP model personalization: inefficient algorithms (based on the exponential mechanism [35]) that establish information-theoretic upper bounds on achievable error, and efficient ones based on popular iterative approaches to non-private personalization [40, 26, 51, 52]. These latter approaches are popular for their convergence speed and low communication overhead. As is often the case, those same features make them attractive starting points for DP algorithms.

**Problem Setting:** Consider a set of $n$ users, and suppose each user $j \in [n]$ holds a data set of $m$ records $D_j = \{(\mathbf{x}_{ij}, y_{ij})\}_{i \in [m]}$ where $\mathbf{x}_{ij} \in \mathbb{R}^d$, $y_{ij} \in \mathbb{R}$. The goal is to learn a personalized model $f_j(\cdot) = f(\cdot; \theta_j) : \mathbb{R}^d \to \mathbb{R}$ for each user $j$, where $\theta_j$ is a vector of parameters describing the model.

We aim to learn a shared, low-dimensional representation for the features that allows users to train good predictors individually. For concreteness, we consider a linear embedding specified by a $d \times k$ matrix $\boldsymbol{U}$, where $k \ll d$. We may think of $\boldsymbol{U}$ either as providing a $k$-dimensional representation of the feature $\mathbf{x}_{ij}$ (as $\boldsymbol{U}^\top \mathbf{x}_{ij}$) or, alternatively, as a compact way to specify a $d$-dimensional regression vector $\theta_j = \boldsymbol{U} \boldsymbol{v}_j$ where $\boldsymbol{v}_j$ is vector of length $k$. In both cases, user $j$'s final predictor has the form

$$f_j(\mathbf{x}_{ij}) = f'(\langle \mathbf{x}_{ij}, \boldsymbol{U} \boldsymbol{v}_j \rangle) = f'(\langle \boldsymbol{U}^\top \mathbf{x}_{ij}, \boldsymbol{v}_j \rangle)$$

One may view this as a model as a two-layer neural network, where the first layer is shared across all users and the second layer is trained individually. A useful setting to have in mind is one where $k \ll m \ll d$—so users do not have enough data to find a good solution on their own, but they do have enough data to find the best vector $\boldsymbol{v}_j$ once an embedding $\boldsymbol{U}$ has been specified. Without loss of generality, we assume $\boldsymbol{U} \in \mathbb{R}^{d \times k}$ to be an orthonormal basis and refer to it as *embedding matrix*. For brevity, we will define the matrix $\boldsymbol{V} = [\boldsymbol{v}_1 | \cdots | \boldsymbol{v}_n] \in \mathcal{C} \subseteq \mathbb{R}^{k \times n}$ with $\boldsymbol{v}_j$s as columns.

**Measure of Accuracy:** Let $\mathcal{L}_{\text{Pop}}(\boldsymbol{U}; \boldsymbol{V}) = \mathbb{E}_{(i,j) \sim_u [m] \times [n], (\mathbf{x}_{ij}, y_{ij}) \sim P_j} \left[ \ell \left( \langle \boldsymbol{U}^\top \mathbf{x}_{ij}, \boldsymbol{v}_j \rangle; y_{ij} \right) \right]$, where the loss function takes the form $\ell : \mathbb{R} \times \mathbb{R} \to \mathbb{R}$. We will focus on excess population risk defined in (1). The privately learned models are denoted by $\left( \boldsymbol{U}^{\text{priv}}, \boldsymbol{V}^{\text{priv}} \right)$. The error measures are defined with respect to any fixed choice of parameters $(\boldsymbol{U}^*, \boldsymbol{V}^*)$.

$$\text{Risk}_{\text{Pop}}\left( \left( \boldsymbol{U}^{\text{priv}}, \boldsymbol{V}^{\text{priv}} \right); (\boldsymbol{U}^*, \boldsymbol{V}^*) \right) = \mathcal{L}_{\text{Pop}}(\boldsymbol{U}^{\text{priv}}, \boldsymbol{V}^{\text{priv}}) - \mathcal{L}_{\text{Pop}}(\boldsymbol{U}^*, \boldsymbol{V}^*). \tag{1}$$

**Alternating Minimization Framework:** We develop an efficient framework based on *alternating minimization* [46, 29, 23]: starting from an initial embedding map $\boldsymbol{U}_0$, the algorithm proceeds in

rounds that alternate between users individually selecting the model $\boldsymbol{v}_j^{(t)}$ that minimizes the error of the predictor $f'(\langle \cdot, \boldsymbol{U}^{(t)} \boldsymbol{v}_j^{(t)} \rangle)$, and then running a DP algorithm, for which user $j$ provides inputs $D_j, \boldsymbol{v}_j^{(t)}$, to privately select a new embedding $\boldsymbol{U}^{(t+1)}$ that minimizes the error of the predictor $f'(\langle \cdot, \boldsymbol{U}^{(t+1)} \boldsymbol{v}_j^{(t)} \rangle)$. In both steps, the optimization to be performed is convex when the loss being optimized is convex. This helps us handle the inherent non-convexity in the problem formulation.

**Instantiation and Analysis for Linear Regression with Gaussian Data:** For the specific case of linear regression with the squared error loss, we show that our framework can be fully instantiated with an efficient algorithm which converges quickly to an optimal solution. For simplicity, we consider the case where the feature vectors and field noise are normally distributed and independent of each user's "true" model $\theta_j^*$, and furthermore that the $\theta_j^*$ vectors admit a common low-dimensional representation $\boldsymbol{U}^* \in \mathbb{R}^{d \times k}$, so that $\theta_j^* = \boldsymbol{U}^* \boldsymbol{v}_j^*$. We show that careful initialization of $\boldsymbol{U}_0$ followed by alternating minimization converges to a near-optimal embedding as long as $m = \omega(k^2)$ and $n = \omega\left(\frac{k^{2.5} d^{1.5}}{\varepsilon}\right)$. Notice that non-privately, one would require $n = \omega(dk)$ users to get any reasonable test error. For standard *private* linear regression in $dk$ dimensions, current state-of-the-art results (Theorem 3.2, [3]) have a sample complexity similar to what we achieve.

**Theorem 1.1** (Special case of Theorem 4.2). *Suppose the output for point $\mathbf{x}_{ij} \sim \mathcal{N}(0, 1)^d$ of user $j$ is given by $y_{ij} \sim \langle (\boldsymbol{U}^*)^\top \mathbf{x}_{ij}, \boldsymbol{v}_j^* \rangle + \mathcal{N}(0, \sigma_F^2)$ where $\boldsymbol{U}^* \in \mathbb{R}^{d \times k}$ is an orthonormal matrix that describes the shared representation, and suppose $\boldsymbol{v}_j^* \sim \mathcal{N}(0, 1)^k$. Let $\sigma_F \leq \sqrt{k}$ and $\varepsilon \leq 1$. Then, assuming the number of users $n$ is at least $(kd)^{1.5}/\varepsilon$, and the number of points per user $m$ is at least $k^2$, with high probability Algorithm 1 learns an embedding matrix $\boldsymbol{U}^{priv}$ such that the average test error of a linear regressor learned over points embedded by $\boldsymbol{U}^{priv}$ is at most $\widetilde{O}\left(\frac{d^3 k^5}{\varepsilon^2 n^2} + \sigma_F^2 \cdot \frac{k}{m}\right)$.*

Our instantiation of the framework in this case has two major components: The initial embedding $\boldsymbol{U}_0$ is derived from users' data by a single noisy averaging step which roughly approximates the $d \times d$ projector onto the $k$-dimensional column space of $\boldsymbol{U}^*$. The idea is that given two data points $(\mathbf{x}_{ij}, y_{ij})$ and $(\mathbf{x}_{(i+1)j}, y_{(i+1)j})$, the expected value of the rank-one matrix $y_{ij} y_{(i+1)j} \mathbf{x}_{ij} \mathbf{x}_{(i+1)j}^\top$ is (when rescaled) a projector onto the space spanned by the regression vector $\theta_j$. Adding these rank-one matrices across many data points and users produces a matrix with high overlap with the desired projector $\boldsymbol{U}^* (\boldsymbol{U}^*)^\top$. This is similar to the approach taken by [13] to design a non-private algorithm for a related, less general setting.

The DP minimization step, which fixes the $\boldsymbol{v}_j$'s and seeks a near-minimal $\boldsymbol{U}$, can be performed using any DP algorithm for convex minimization [9, 4]. In this particular case, one can view this step as solving a linear regression problem in which $\boldsymbol{U}$ represents a list of $dk$ real parameters: once $\mathbf{x}$ and $\boldsymbol{v}$ are fixed, $\langle \boldsymbol{U}^\top \mathbf{x}, \boldsymbol{v} \rangle = \mathbf{x}^\top \boldsymbol{U} \boldsymbol{v}$ is a linear function of $\boldsymbol{U}$.

For the analysis to be tractable, we restrict our attention to linear regression with independent, normally-distributed features. However, the framework we provide is more general, and can be applied to a wider class of models. Developing mathematical tools to analyse the behavior of noisy alternating minimization algorithms in more general settings remains an important open question.

Additionally, we run simulations on synthetic data to demonstrate the effectiveness of our proposed algorithm. Our algorithm reaches a significantly better privacy-utility tradeoff compared to two baselines: i) each user uses their own data, and ii) all users jointly learn a single model under differential privacy.

**Information-theoretic Upper Bounds:** In addition to developing efficient algorithms for particular settings, we give upper bounds on the achievable error of user-level DP model personalization via inefficient algorithms. Specifically, we consider the natural approach of using the exponential mechanism [35] to select a common structure that provides low prediction error on average across users. For the specific case of a shared linear embedding (a generalization of the linear regression setting above), when the feature vectors are drawn i.i.d. from $\mathcal{N}(0, 1)^d$, and when the $\boldsymbol{v}_j^*$'s are drawn i.i.d. from $\mathcal{N}(0, 1)^k$, we provide an upper bound showing that $n = \omega\left(\frac{k^{1.5} d^{1.5}}{\varepsilon}\right)$ users suffice to learn a good model, assuming $m$ is sufficiently large for users to train the remaining parameters locally

(Theorem 5.2). In comparison to alternating minimization, the sample complexity is better by a factor of $k$.

In summary, we initiate a systematic study of differentially private model personalization in the practically important few-shot (or per-user sparse data) learning regime. We propose using users' data to learn a strong common representation/embedding using differential privacy, that can in turn be used to learn sample efficient models for each user. Using a simple but foundational problem setting, we demonstrate rigorously that this technique can indeed learn accurate common representation as well as personalized models, despite users housing only a small number of data points.

## 1.2 Related Work

**Personalization Frameworks:** Model personalization is a special case of multitask or few-shot learning [10, 25] where the goal is to leverage shared structure amongst multiple tasks to better learn the individual tasks. There are many different frameworks for multi-task learning, each capturing a different kind of shared structure. In the context of model personalization, where tasks correspond to users, two broad approaches stand out.

*"Neighboring models".* This approach assumes that while each user learns their own model, all or a fraction of the models are close to each other thus can be learned together [18, 25].

*"Common representation".* This approach, which we adopt in this paper, assumes a low-dimensional shared subspace where all points can be represented and now each user/task can learn a sample efficient model to solve the individual task [47, 38]. A common instantiation is a DNN architecture in which the weights in the last layer are user-specific but other weights are shared. Algorithmically, this second approach is more complex since it entails simultaneously finding an accurate representation of data and models building upon those representations. But several studies [38, 47] have shown it to be significantly more effective than other approaches like neighboring models.

Recent works on this approach (e.g. [44, 47, 22, 40]) follow a similar training strategy to ours— that is, they alternatively update the shared representation using gradient descent and then finetune individual classifiers [38, 30, 47]. In particular, the Almost-No-Inner-Loop (ANIL) method by [38] is most similar to the alternating optimization method that we adopt (see Algorithm 1). Theoretical understanding of these methods generally lag significantly behind their empirical success. However, several interesting recent results explain the effectiveness of these methods on simple tasks [13, 46]. Most of the papers in this domain focus on the linear regression problem with a shared low-dimensional representation that we study [46, 11, 48]. They show that one can provide much better estimates for the shared representation, and overall prediction error, by pooling information than would be possible for individual users acting alone. These existing analyses do not allow for noise in the iterations. In fact, for the general problem, the noise can lead to suboptimal solutions. Thus, a key contribution of our work is to show that in a widely studied setting, alternating minimization converges even when the minimization of $U$ is noisy.

**Privacy:** In our setting, the data set is made up of users' individual data sets $D_1, ..., D_n$, where each $D_j$ potentially contains many records (labeled training examples). Users interact via a central algorithm, which we assume for simplicity to be implemented correctly and securely (either by a trusted party or using cryptographic techniques like multiparty computation). This algorithm provides output to each of the users. We aim to control what those outputs leak about the users' input data.

That is, presence/absence of user and its entire data should not affect the outputs significantly. This notion is known as *user-level* or task-level privacy and has been widely studied in the literature [34, 31], albeit mostly without personalization component. The only works we are aware of that look at personalization (or multitask learning more generally) with user-level guarantees are [19] and [24]. Geyer et al. [19] consider the "neighboring models" approach, which cannot work in the setting we study. Jain et al. [24] consider matrix completion, which can be viewed as a version of our setting in which training examples are limited to indicator vectors (items from a known discrete set).

A few studies attempt to provide only *record-level* privacy – a significantly weaker notion of privacy where presence/absence of only single record should be undetected by the output of the model. While the notion has been studied extensively for the standard non-personalized models [27, 9], for personalized models the literature is somewhat limited [21, 32]. The work of [32] discusses both task- and record-level privacy, but ultimately provides only algorithms that satisfy the weaker guarantee.

As mentioned above, our goal is to provide strong user-level privacy guarantees so such methods do not apply in our case.

## 1.3 Notation

We denote all matrices with bold upper case letters (e.g., $\boldsymbol{A}$), and all vectors with bold lower case letters ($\boldsymbol{a}$). Unless specified explicitly, all vectors are column vectors. We denote the clipping operation on a vector $\boldsymbol{a}$ as $\mathsf{clip}(\boldsymbol{a}; \zeta) = \boldsymbol{a} \cdot \min\left\{1, \frac{\zeta}{\|\boldsymbol{a}\|_2}\right\}$.

## 2 Background on Privacy

**Billboard model:** In this paper, we operate in the billboard model [20] of differential privacy [15, 14, 36]. Consider $n$ users, and a computing server. The server runs a differentially private algorithm on sensitive information from the users, and broadcasts the output to all the users. Each user $j \in [n]$ can then use the broadcasted output in a computation that solely relies on her data. The output of this computation is not made available to other users. A block schematic is shown in Figure 1. One important attribute of the billboard model is that it trivially satisfies joint differential privacy [28].

**User-level privacy protection:** In this work, we provide user-level privacy protection [16]. I.e., from the output of the algorithm available to an adversary, they will not be able to detect the presence/absence of *all the data samples belonging to a single user.* Correspondingly, in the definition of differential privacy below (Definition 2.1), a "record" consists of all the data samples belonging to a single user. Furthermore, we adhere to the replacement model of privacy, where the protection is with respect to the replacement of a user with another, instead of the presence/absence of a user.

**Definition 2.1** (Differential Privacy [15, 14, 36]). *A randomized algorithm $\mathcal{A}$ is $(\varepsilon, \delta)$-differentially private if for any pair of data sets $D$ and $D'$ that differ in one record (i.e., $|D \triangle D'| = 1$), and for all $S$ in the output range of $\mathcal{A}$, we have*

$$\mathbf{Pr}[\mathcal{A}(D) \in S] \leq e^{\varepsilon} \cdot \mathbf{Pr}[\mathcal{A}(D') \in S] + \delta,$$

*where probability is over the randomness of $\mathcal{A}$. Similarly, an algorithm $\mathcal{A}$ is $(\alpha, \rho)$- Rényi differentially private (RDP) if $D_{\alpha}(\mathcal{A}(D)\|\mathcal{A}(D')) \leq \rho$, where $D_{\alpha}$ is the Rényi divergence of order $\alpha$.*

## 3 Model Personlization via Private Alternating Minimization

In this section, we first provide a generic/meta algorithm for private model personalization (Algorithm 1 (Algorithm $\mathcal{A}_{\mathsf{Priv-AltMin}}$)). The main idea is to alternate between two states for $T$ iterations, i.e., for $t \in [T]$, (i) Estimate the best embedding matrix $\boldsymbol{U}^{(t)}$ based on the current personalized models $\left[\boldsymbol{v}_1^{(t)}, \ldots, \boldsymbol{v}_n^{(t)}\right]$ while preserving *user-level* $(\alpha, \rho)$-RDP, and (ii) update the personalized modes based on the updated embedding matrix $\boldsymbol{U}^{(t)}$. Finally, output $\boldsymbol{U}^{\mathtt{priv}} \leftarrow \boldsymbol{U}^{(T+1)}$, which will be used by each user $j \in [n]$ to train her final personalized model $\boldsymbol{v}_j^{\mathtt{priv}}$. While Algorithm $\mathcal{A}_{\mathsf{Priv-AltMin}}$ is a fairly natural method for model personalization, to the best of our knowledge, this is the first work that formally studies the privacy/utility trade-offs under user-level privacy. Prior works [40, 37] have used similar ideas in the *non-private* meta-learning setting. The estimation of the embedding matrix can be implemented by

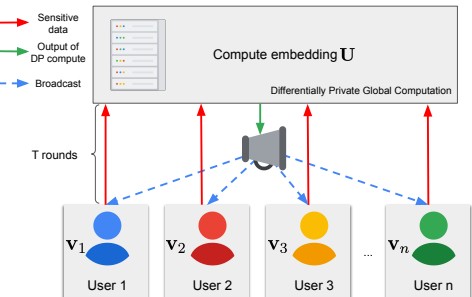

Figure 1: User-compute interaction in the billboard model. Shaded boxes represent privileged computation. $\boldsymbol{U}$ refer to the common embedding function, and $\boldsymbol{v}_j$ refers to the model for user $j \in [n]$.

any differentially private convex optimization algorithm (e.g., DP-SGD [42, 4, 1]). As discussed in Section 4, for specific case of linear regression, we can perturb the sufficient statistics to obtain differential privacy guarantee, and then optimize over it. A similar idea was used in [41, 39].

We provide a formal description in Algorithm 1. In Section 4, we instantiate it in the context of personalized linear regression. There, we also provide formal excess population risk guarantees under

---

**Algorithm 1** $\mathcal{A}_{\text{Priv-AltMin}}$: Differentially Private Alternating Minimization Meta-algorithm

---

**Require:** Data sets from each user $j \in [n]$: $D_j = \{(\mathbf{x}_{ij} \in \mathbb{R}^d, y_{ij} \in \mathbb{R}) : i \in [m]\}$ for $m$ mod $4 = 0$, rank of the projector: $k$, privacy parameters: $(\alpha, \rho)$, number of iterations: $T$, initial rank-$k$ subspace matrix: $\boldsymbol{U}^{\text{init}}$, loss function: $\ell$.

1: Initialize $\boldsymbol{U}^{(1)} \leftarrow \boldsymbol{U}^{\text{init}}$.
2: Randomly permute the users $j \in [n]$ via permutation $\pi \sim_{\text{unif}} [n]$. Set $j \leftarrow \pi(j), \forall j \in [n]$.
3: **for** $t \in [T]$ **do**
4:     $\mathcal{S}_t \leftarrow \left[ 1 + \lceil \frac{(t-1)n}{T} \rceil, \lceil \frac{tn}{T} \rceil \right]$.
5:     Each user $j \in [\mathcal{S}_t]$ independently solves $\boldsymbol{v}_j^{(t)} \leftarrow \underset{\|\boldsymbol{v}\|_2 \leq \mathbb{R}^k}{\arg\min} \frac{4}{m} \sum_{i \in [m/4]} \ell \left( \langle (\boldsymbol{U}^{(t)})^\top \mathbf{x}_{ij}, \boldsymbol{v} \rangle; y_{ij} \right)$.
6:     Estimate $\boldsymbol{U}^{(t+1)} \leftarrow \underset{\boldsymbol{U} \in \mathcal{K}}{\arg\min} \frac{4}{m \cdot |\mathcal{S}_t|} \sum_{i \in [m/4+1, m/2], j \in \mathcal{S}_t} \ell \left( \langle \boldsymbol{U}^\top \mathbf{x}_{ij}, \boldsymbol{v}_j^{(t)} \rangle; y_{ij} \right)$ under $(\alpha, \rho)$-

    RDP, where $\mathcal{K}$ is the set of all rank-$k$ matrices with orthonormal columns in $\mathbb{R}^{d \times k}$.
7: **end for**
8: $\boldsymbol{U}^{\text{priv}} \leftarrow \boldsymbol{U}^{(T+1)}$.

---

some data generating assumption. Since Line 6 guarantees $(\alpha, \rho)$-RDP, and disjoint sets of users are used in each iteration, we can conclude that the whole algorithm guarantees $(\alpha, \rho)$-RDP.

# 4 Instantiating Algorithm $\mathcal{A}_{\text{Priv-AltMin}}$ with Linear Regression

In this section, we instantiate Algorithm $\mathcal{A}_{\text{Priv-AltMin}}$ (Algorithm 1) in the context of linear regression. While our privacy guarantees hold for any instantiation of the training data, the utility guarantees hold under the following data generating assumption.

**Data generation:** We instantiate the problem description in Section 1.1 as follows. There is a fixed model $\boldsymbol{v}_j^* \in \mathbb{R}^k$ for each user $j \in [n]$, and a fixed rank-$k$ matrix with orthonormal columns $\boldsymbol{U}^* \in \mathbb{R}^{d \times k}$ across all users. Let $\boldsymbol{V}^* := [\boldsymbol{v}_1^* | \cdots | \boldsymbol{v}_n^*]$. For each feature vector $\mathbf{x}_{ij} \in \mathbb{R}^d$, the response $y_{ij}$ is given by:

$$ y_{ij} = \langle (\boldsymbol{U}^*)^\top \mathbf{x}_{ij}, \boldsymbol{v}_j^* \rangle + \boldsymbol{z}_{ij}, \ \ \boldsymbol{z}_{ij} \sim \mathcal{N}(0, \sigma_{\mathsf{F}}^2). \tag{2} $$

In Theorem 4.2, we provide the privacy and utility guarantee for an instantiation of Algorithm 1 (Algorithm $\mathcal{A}_{\text{Priv-AltMin}}$) where the loss function is $\ell \left( \langle \boldsymbol{U}^\top \mathbf{x}_{ij}, \boldsymbol{v} \rangle; y_{ij} \right) = \left( y_{ij} - \langle \boldsymbol{U}^\top \mathbf{x}_{ij}, \boldsymbol{v} \rangle \right)^2$. We will adhere to Assumptions 4.1 for the utility analysis.

**Assumption 4.1** (Assumptions for Utility Analysis). *Let* $\lambda_i > 0$ *be the $i$-th eigenvalue of* $\frac{1}{n} \left( \boldsymbol{V}^* (\boldsymbol{V}^*)^\top \right)$, *and let* $\mu := \max_{j \in [n]} \|\boldsymbol{v}_j^*\|_2 / \sqrt{k \lambda_k}$ *be the incoherence parameter. Let Noise-to-signal ratio be* $\text{NSR} = \frac{\sigma_{\mathsf{F}}}{\sqrt{\lambda_k}}$. *We assume: (i)* $\forall i \in [m], j \in [n], \mathbf{x}_{ij} \sim_{\text{iid}} \mathcal{N}(0,1)^d$, *and corresponding $y_{ij}$ be generated using (2), (ii)* $m = \widetilde{\Omega} \left( (1 + \text{NSR}) \cdot k + k^2 \right)$, *(iii)* $n = \widetilde{\Omega} \left( \frac{\lambda_1}{\lambda_k} \cdot \mu^2 dk + \left( \frac{\lambda_1}{\lambda_k} \right)^2 \frac{d}{k^2} \cdot \left( \text{NSR}^2 + \mu^2 k \right)^2 + \frac{\lambda_1}{\lambda_k} \cdot \Delta_{(\varepsilon, \delta)} \cdot \left( \text{NSR}^2 + \mu^2 k \right) d^{3/2} \right)$. *Here,* $\widetilde{\Omega}(\cdot)$ *hides* $\text{polylog}(n, m, k)$.

**Theorem 4.2** (Main Result. Bound on Excess Risk). *Let* $\boldsymbol{V}^{priv} = [\boldsymbol{v}_1^{priv}, \ldots, \boldsymbol{v}_n^{priv}]$ *with*

$$ \boldsymbol{v}_j^{priv} \leftarrow \underset{\boldsymbol{v} \in \mathbb{R}^k}{\arg\min} \frac{2}{m} \sum_{\frac{m}{2} < i \leq m} \left( y_{ij} - \langle (\boldsymbol{U}^{priv})^\top \mathbf{x}_{ij}, \boldsymbol{v} \rangle \right)^2. $$

*Let Assumption 4.1 hold. Then, Algorithm $\mathcal{A}_{\text{Priv-AltMin}}$ with parameters in Lemma 4.4 and $\Delta_{(\varepsilon, \delta)} := \frac{\sqrt{16 \log(1/\delta)}}{\varepsilon}$ outputs $\boldsymbol{U}^{priv}$ such that i) it is $(\varepsilon, \delta)$-differentially private, and ii) it has the following excess population risk w.p. at least $1 - 1/n^9$ (over the randomness of data generation and the*

---

**Algorithm 2** Instantiating Line 6 of Algorithm 1 ( Algorithm $\mathcal{A}_{\mathsf{Priv\text{-}AltMin}}$)

---

**Require:** Set of users at time step $t \in [T]$: $\mathcal{S}_t$. Current models: $\left\{\boldsymbol{v}_j^{(t)} : j \in \mathcal{S}_t\right\}$, data samples: $\{(\mathbf{x}_{ij}, y_{ij}) : j \in \mathcal{S}_t, i \in [m]\}$, privacy parameter: $\Delta_{(\varepsilon, \delta)}$, clipping threshold for model: $\eta$, clipping threshold for response: $\zeta$.

1: $\boldsymbol{W}_{ij} = \mathsf{clip}\left(\overrightarrow{\mathbf{x}_{ij}\boldsymbol{v}_j^\top}; \eta\right)$ and $\widetilde{y}_{ij} = \mathsf{clip}\left(y_{ij}; \zeta\right)$ for all $i \in [m/4+1, m/2], j \in \mathcal{S}_t$.

2: $\boldsymbol{W}_{\mathsf{priv}} \leftarrow \displaystyle\sum_{j \in \mathcal{S}_t, i \in [m/4+1, m/2]} \boldsymbol{W}_{ij}\boldsymbol{W}_{ij}^\top + \mathcal{N}_{\mathsf{sym}}\left(0, m^2\eta^4\Delta_{(\varepsilon,\delta)}^2/4\right)^{dk \times dk}$, and

   $\boldsymbol{b}_{\mathsf{priv}} \leftarrow \displaystyle\sum_{j \in \mathcal{S}_t, i \in [m/4+1, m/2]} \widetilde{y}_{ij}\boldsymbol{W}_{ij} + \mathcal{N}\left(0, m^2\zeta^2\eta^2\Delta_{(\varepsilon,\delta)}^2/4\right)^{dk}$

3: $\overrightarrow{\boldsymbol{Z}}^{(t+1)} \leftarrow \underset{\boldsymbol{u} \in \mathbb{R}^{dk}}{\arg\min} \frac{4}{m \cdot |\mathcal{S}_t|}\left(\boldsymbol{u}^\top \boldsymbol{W}_{\mathsf{priv}}\boldsymbol{u} - 2\boldsymbol{u}^\top \boldsymbol{b}_{\mathsf{priv}}\right)$

4: **return** $\boldsymbol{U}^{(t+1)} \leftarrow Q$ part of the $QR$-decomposition of $\boldsymbol{Z}^{(t+1)}$

---

*algorithm):*

$$\mathsf{Risk}_{Pop}\left(\left(\boldsymbol{U}^{priv}, \boldsymbol{V}^{priv}\right); (\boldsymbol{U}^*, \boldsymbol{V}^*)\right) \leq$$
$$= O\left(\frac{\Delta_{(\varepsilon,\delta)}(\sigma_{\mathrm{F}}^2 + \mu^2 k^2 d\lambda_k)(\mu^4 k^3 d^2)}{n^2} + \frac{\sigma_{\mathrm{F}}^2 \mu^4 k^2 d}{nm}\right) \cdot \mathrm{polylog}\,(d, n) + \frac{k}{m} \cdot \sigma_{\mathrm{F}}^2.$$

See supplementary material for the proof.

**Remark 1.** Let us understand the bound above for a simple setting where the personal model for each user $\boldsymbol{v}_j^* \sim \mathcal{N}(0,1)^k$. Assuming large enough $n$, this implies that $\lambda_k \approx 1$ and $\mu \approx \widetilde{O}(1)$. Now even when $\boldsymbol{V}^*$ is *known a priori*, to obtain a reasonable estimate of $\boldsymbol{U}^*$, we need to solve the following linear regression problem while ensuring DP: $\boldsymbol{U}^{priv} = \min_{\boldsymbol{U}} \sum_{ij}(y_{ij} - \langle \mathbf{x}_{ij}(\boldsymbol{v}_j^*)^\top, \boldsymbol{U}\rangle)^2$. Note that $\mathbf{x}_{ij}(\boldsymbol{v}_j^*)^\top$ is isotropic. Now, *without* differential privacy, the information theoretical optimal estimation error is $\Theta\left(\sigma_{\mathrm{F}}^2 \cdot \frac{dk}{nm}\right)$, where $dk$ is the size of the linear regression problem and $mn$ is the number of samples. Now, if we were to solve the above regression problem with DP, the best known algorithm [41] will have an additional error of $\widetilde{O}\left(\left(\kappa \cdot \frac{dk}{n\varepsilon}\right)^2\right)$, where $\kappa = \sigma_{\mathrm{F}} + \max_{ij}\|\mathbf{x}_{ij}(\boldsymbol{v}_j^*)^\top\|_F \cdot \|\boldsymbol{U}^*\|_F = \widetilde{O}(\sigma_{\mathrm{F}} + \sqrt{dk^2})$. Note that the first two terms in Theorem 4.2 indeed match $O\left(\left(\kappa \cdot \frac{dk}{n\varepsilon}\right)^2 + \sigma_{\mathrm{F}}^2 \cdot \frac{dk}{nm}\right)$ up to an additional factor of $k$ and up to $\mathrm{polylog}\,(d, n)$ factors. Finally, the last error term in the above theorem is due to excess risk in estimating $\boldsymbol{v}^*$ for a given user with $m$ samples, and is information theoretically optimal.

**Remark 2.** Under the assumption in Remark 1 and for $\sigma_{\mathrm{F}} = 0$, the sample complexity for Theorem 4.2 is $n = \widetilde{\omega}(k^{2.5}d^{1.5}/\varepsilon + d)$ and $m = \widetilde{\omega}(k^2)$. Note that, for $\varepsilon \to \infty$, the complexity is $O(k)$ worse than the information theoretic optimal. Furthermore, the sample complexity suffers from an additional $\sqrt{d}$ for constant $\varepsilon$ compared to non-private case. Even for standard linear regression, a similar additional $\sqrt{d}$ factor is present in the sample complexity bound [41]; we leave further investigation into the optimal sample complexity for future work.

In Section 4.1, we show an instantiation of Algorithm $\mathcal{A}_{\mathsf{Priv\text{-}AltMin}}$ (Algorithm 1) s.t. if the embedding matrix ($\boldsymbol{U}^{\mathtt{init}}$) is initialized well, then $\boldsymbol{U}^{\mathtt{priv}}\left(\boldsymbol{U}^{\mathtt{priv}}\right)^\top$ converges in $\|\cdot\|_F$ to $\boldsymbol{U}^*(\boldsymbol{U}^*)^\top$. In Section 4.2, we provide an algorithm to obtain a good initialization of the embedding matrix ($\boldsymbol{U}^{\mathtt{init}}$). Combining these two results imply Theorem 4.2.

### 4.1   Local Subspace Convergence

In Algorithm 2, we instantiate Line 6 of Algorithm $\mathcal{A}_{\mathsf{Priv\text{-}AltMin}}$. For any matrix $\boldsymbol{A} \in \mathbb{R}^{d_1 \times d_2}$, let $\overrightarrow{\boldsymbol{A}} \in \mathbb{R}^{d_1 d_2}$ be the vectorized representation with columns of $\boldsymbol{A}$ placed consecutively. Let $\mathcal{N}_{\mathsf{sym}}(0, \sigma^2)^{d \times d}$ denote a Wigner matrix with entries drawn i.i.d. from $\mathcal{N}(0, \sigma^2)$. The privacy guarantee of Algorithm 2 is presented in Lemma 4.3 and the local subspace guarantee in Lemma 4.4.

**Lemma 4.3** (Privacy guarantee). *If we set* $\Delta_{(\varepsilon,\delta)} = \sqrt{8\log(1/\delta)}/\varepsilon$, *then instantiation of Algorithm* $\mathcal{A}_{\mathsf{Priv\text{-}AltMin}}$ *with Algorithm 2 is* $(\varepsilon, \delta)$-*differentially private in the billboard model.*

**Lemma 4.4** (Local Subspace Convergence). *Recall Assumptions 4.1. In Algorithm 2, let model clipping threshold $\eta = \widetilde{O}(\mu\sqrt{\lambda_k dk})$, and response clipping threshold $\zeta = \widetilde{O}\left(\sigma_F + \mu\sqrt{k\lambda_k}\right)$. Let the number of iterations of Algorithm 1 (Algorithm $\mathcal{A}_{\mathsf{Priv\text{-}AltMin}}$) be $T = \Omega\left(\log\left(\frac{(\lambda_1/\lambda_k)}{NSR + \Delta_{(\varepsilon,\delta)}}\right)\right)$. Finally, assume $U^{init}$ be s.t. $\|(\mathbb{I} - U^*(U^*)^\top)U^{init}\|_F \leq \frac{\lambda_k}{32\lambda_1}$. We have the following for Algorithm 1 (Algorithm $\mathcal{A}_{\mathsf{Priv\text{-}AltMin}}$), instantiated with Algorithm 2, w.p. at least $1 - 1/n^{10}$ (over the randomness of data generation and the algorithm):*

$$\left\|\left(\mathbb{I} - U^*(U^*)^\top\right)U^{priv}\right\|_F = \widetilde{O}\left(\frac{\Delta_{(\varepsilon,\delta)}(NSR + \mu\sqrt{dk^2})\mu\sqrt{k^2d^2}}{n} + \frac{NSR \cdot \mu\sqrt{kd}}{\sqrt{nm}}\right).$$

*Here, the noise-to-signal-ratio $NSR = \frac{\sigma_F}{\sqrt{\lambda_k}}$ and privacy parameter $\Delta_{(\varepsilon,\delta)} = \frac{\sqrt{8\log(1/\delta)}}{\varepsilon}$. In $\widetilde{O}(\cdot)$, we hide* polylog $(d, n)$.

See supplementary material for the proofs. The analysis of Lemma 4.4 roughly follows the analysis of alternating minimization [46], while accounting for the noise introduced due to privacy. At each iteration, we show that the embedding subspace gets closer in the Frobenius norm, and each of the personalized models gets closer in the $\ell_2$-norm.

## 4.2 Initialization Algorithm

In Algorithm 3, we describe a private estimator for the estimation of $U^*$. This estimator eventually gets used in initializing the linear regression instantiation of Algorithm 1. We provide the privacy and subspace closeness guarantees in Lemma 4.5 and 4.6, with proofs in supplementary material.

---

**Algorithm 3** $\mathcal{A}_{\mathsf{Priv\text{-}init}}$: Private Initialization Algorithm for Algorithm $\mathcal{A}_{\mathsf{Priv\text{-}AltMin}}$

---

**Require:** Data sets from each user $j \in [n]$: $D_j = \{(\mathbf{x}_{ij} \in \mathbb{R}^d, y_{ij} \in \mathbb{R}) : i \in [m]\}$, clipping bound for response: $\zeta$, noise standard dev. for privacy: $\Delta_{(\varepsilon,\delta)}$, and rank of the orthonormal basis: $k$.

1: $W_{ij} \leftarrow \mathsf{sym}\left(\frac{\mathbf{x}_{(2i)j}\mathbf{x}_{(2i+1)j}^\top}{\|\mathbf{x}_{(2i)j}\|_2 \cdot \|\mathbf{x}_{(2i+1)j}\|_2} \cdot \mathsf{clip}\left(y_{(2i)j}; \zeta\right) \cdot \mathsf{clip}\left(y_{(2i+1)j}; \zeta\right)\right)$ for all $i \in [m/2]$ and $j \in [n]$. Here, $\mathsf{sym}(W)$ makes a matrix $\in \mathbb{R}^{d\times d}$ symmetric by replicating the upper triangle.

2: $M^{\mathsf{Noisy}} \leftarrow \frac{2}{nm}\left(\sum_{i \in [m/2], j \in [n]} W_{ij} + \mathcal{N}_{\mathsf{sym}}\left(0, \Delta_{(\varepsilon,\delta)}^2\zeta^4 m^2\right)^{d\times d}\right)$.

3: $U^{priv} \leftarrow$ Top-$k$ eigenvectors of $M^{\mathsf{Noisy}}$ as columns.

---

**Lemma 4.5** (Privacy guarantee). *If we set $\Delta_{(\varepsilon,\delta)} = \sqrt{8\log(1/\delta)}/\varepsilon$, Algorithm 3 (Algorithm $\mathcal{A}_{\mathsf{Priv\text{-}init}}$) is $(\varepsilon, \delta)$-differentially private.*

**Lemma 4.6** (Subspace closeness). *Recall Assumptions 4.1. Let the clipping bound for response be $\zeta = \widetilde{O}(\sigma_F + \mu\sqrt{k\lambda_k})$. We have the following for Algorithm 3 (Algorithm $\mathcal{A}_{\mathsf{Priv\text{-}init}}$) w.p. at least $1 - 1/n^{10}$:*

$$\left\|\left(\mathbb{I} - U^*(U^*)^\top\right)U^{priv}\right\|_2 = \widetilde{O}\left(\frac{\Delta_{(\varepsilon,\delta)}\left(NSR^2 + \mu^2 k\right)d^{3/2}}{n} + \frac{(NSR^2 + \mu^2 k)\sqrt{d}}{\sqrt{nm}}\right).$$

*Here, privacy parameter $\Delta_{(\varepsilon,\delta)} = \frac{\sqrt{8\log(1/\delta)}}{\varepsilon}$. In $\widetilde{O}(\cdot)$, we hide* polylog $(d, n)$.

The proof goes via direct analysis of the distance between the estimated subspace from the training examples, and the true subspace. While the convergence guarantee in Lemma 4.6 is unconditional, it is weaker than Lemma 4.4, especially in its dependence on $k$ and NSR.

Lemma 4.6 implies that under Assumption 4.1, $\left\|\left(\mathbb{I} - U^*(U^*)^\top\right)U^{priv}\right\|_F = O\left(\frac{\lambda_k}{\lambda_1}\right)$, which is sufficient to satisfy the initialization condition in Lemma 4.4. Hence, if we initialize $U$ using Algorithm 3 (Algorithm $\mathcal{A}_{\mathsf{Priv\text{-}init}}$) with a *disjoint* set of samples for each user, it immediately follows that the the local convergence guarantee in Lemma 4.4 is indeed a global convergence guarantee.

---
**Algorithm 4** $\mathcal{A}_{\mathsf{Exp}}$: Joint Differentially Private ERM via Exponential Mechanism
---
**Require:** Data sets from each user $j \in [n]$: $D_j = \{(\mathbf{x}_{ij} \in \mathbb{R}^d, y_{ij} \in \mathbb{R}) : i \in [m]\}$ where $m$ mod $2 = 0$; model $\ell_2$-norm constraint $C$; clipping bound on the projected features $L_f$; privacy parameter $\varepsilon$; rank of the projection matrix $k$; net width $\phi$, loss function $\ell : \mathbb{R} \times \mathbb{R} \to \mathbb{R}$; Lipschitz constant $\xi$ of w.r.t. its first parameter.

1: Define a score function for any rank-$k$ matrix with orthonormal columns $\boldsymbol{U} \in \mathbb{R}^{d \times k}$ as

$$\mathsf{score}\,(\boldsymbol{U}) = \sum_{j \in [n]} \left( \min_{\|\boldsymbol{v}_j\|_2 \leq C} \frac{2}{m} \sum_{i \in [m/2]} \ell\left(\langle \mathsf{clip}\left(\boldsymbol{U}^\top \mathbf{x}_{ij}; L_f\right), \boldsymbol{v}_j\rangle; y_{ij}\right) \right).$$

2: Define a net $\mathcal{N}^\phi$ of $\|\cdot\|_F$-radius $\phi$ over matrices with orthonormal columns in $\mathbb{R}^{d \times k}$.

3: Sample $\boldsymbol{U}^{\mathrm{priv}} \in \mathcal{N}^\phi$ with $\mathbf{Pr}[\boldsymbol{U}^{\mathrm{priv}} = \boldsymbol{U}] \propto \exp\left(-\frac{\varepsilon n}{8 L_f C \xi} \cdot \mathsf{score}\,(\boldsymbol{U})\right)$.

4: Each user $j \in [n]$ *independently* estimates $\boldsymbol{v}_j^{\mathrm{priv}} \leftarrow \underset{\|\boldsymbol{v}\|_2 \leq C}{\arg\min} \frac{2}{m} \sum_{i=m/2+1}^{m} \ell\left(\langle (\boldsymbol{U}^{\mathrm{priv}})^\top \mathbf{x}_{ij}, \boldsymbol{v}\rangle; y_{ij}\right)$.

---

## 5 Exponential Mechanism based Model Personalization

In this section, we take a more general approach towards outputting a projector $\boldsymbol{U}^{\mathrm{priv}}$ that approximately minimizes the excess population risk without worrying about actually estimating the projector onto $\boldsymbol{U}^*$. Here, as we only care about low-excess risk, as opposed to subspace closeness, we can guarantee better convergence under milder assumptions. Recall the loss function $\mathcal{L}_{\mathrm{Pop}}(\boldsymbol{U}, \boldsymbol{V})$ from (1).

We want to optimize $\min_{\boldsymbol{U} \in \mathcal{K}} \left( \min_{\boldsymbol{V} \in \mathbb{R}^{d \times n}, \|\boldsymbol{v}_j\|_2 \leq C} \mathcal{L}_{\mathrm{Pop}}(\boldsymbol{U}, \boldsymbol{V}) \right)$ while ensuring $\varepsilon$-DP in the billboard model. (Here $\mathcal{K} \in \mathbb{R}^{d \times k}$ is the set of matrices with orthonormal columns, and $\boldsymbol{v}_j$ corresponds to the $j$-th column of $\boldsymbol{V}$.) To that end, we will use the exponential mechanism [35], over an $\ell_F$-net of radius $\phi$ over $\mathcal{K}$. The algorithm is presented in Algorithm 4 (Algorithm $\mathcal{A}_{\mathsf{Exp}}$).

The privacy analysis of Algorithm $\mathcal{A}_{\mathsf{Exp}}$ follows from the standard analysis of exponential mechanism, and the utility analysis goes via first proving an excess empirical risk bound, and then appealing to uniform convergence to get to excess population risk bound.

**Theorem 5.1** (Privacy guarantee). *Algorithm 4 is $\varepsilon$-differentially private in the billboard model.*

**Theorem 5.2** (Utility guarantee). *Suppose the loss function $\ell$ is $\xi$-Lipschitz in its first parameter, and $C$ is the bound on the constraint set. Set the net size $\phi = 1/(\varepsilon n)$ and the clipping norm $L_f = 40\sqrt{d} \cdot \log(nm)$ in Algorithm 4. Assuming $\varepsilon \in (0, 1)$ and that the feature vectors are drawn i.i.d. from $\mathcal{N}(0, 1)^d$, we have (w.p. $\geq 1 - 1/\min\{d, n\}^{10}$):*

$$\mathsf{Risk}_{Pop}\left(\left(\boldsymbol{U}^{priv}, \boldsymbol{V}^{priv}\right); (\boldsymbol{U}^*, \boldsymbol{V}^*)\right) = O\left(\xi C \cdot \left(\frac{k \cdot d^{1.5}}{\varepsilon n} + \frac{\sqrt{k}}{\sqrt{m}}\right)\right) \cdot \mathrm{polylog}\,(d, n).$$

*Here, $\boldsymbol{U}^*$ and $\boldsymbol{V}^*$ are any fixed values of the common embedding matrix (orthonormal, in $\mathbb{R}^{d \times k}$) and matrix of individual regression vectors (in $\mathbb{R}^{n \times k}$, with rows of norm at most $C$), respectively.*

See supplementary material for the proofs of Theorems 5.1 and 5.2.

**Comparison of the utility guarantee to Theorem 4.2:** The utility guarantee for Algorithm $\mathcal{A}_{\mathsf{Exp}}$ (Theorem 5.2) is much more general than that in Theorem 4.2. Unlike Theorem 4.2, it allows arbitrary Lipschitz loss function $\ell$, and any distribution over the feature vectors. However, for linear regression with i.i.d. spherical normal feature vectors and setting the diameter of the constraint set $C = \sqrt{k}$, one can make Theorems 4.2 and 5.2 comparable. Theorem 4.2 shows an excess population risk $\widetilde{O}\left(\frac{k^5 d^3}{\varepsilon^2 n^2} + \frac{k}{m}\right)$ whereas Theorem 5.2 gives $\widetilde{O}\left(\frac{\sqrt{k^3 d^3}}{\varepsilon n} + \sqrt{\frac{k^2}{m}}\right)$. Theorem 4.2 is tighter in the regime where $n = \Omega(k^{3.5} d^{1.5}/\varepsilon)$. This difference is comparable to the so-called *fast rates* [43]. However, the sample complexity of Theorem 5.2 is better in terms of $m$ by a factor of $k^{1.5}$.

## 6 Numerical Simulation

In this section, we provide numerical simulations to validate our theoretical results. There are three baselines: i) each user solves their problem using their own data without consulting with other

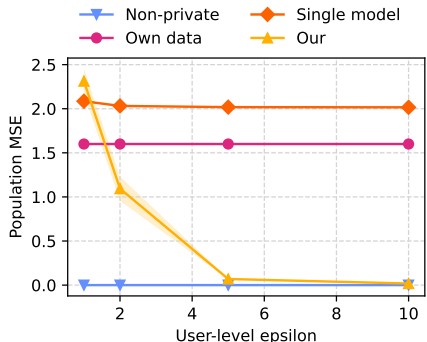

Figure 2: MSE vs. per-user $\varepsilon$ for linear regression.

users (called *own data*), and ii) a single differentially private global model trained on all users data together (called *single model*), and iii) non-private variant of alternating minimization, trained up to convergence. Overall, we observe that Algorithm 1 (Algorithm $\mathcal{A}_{\mathsf{Priv\text{-}AltMin}}$) outperforms the private baselines by a significant margin.

We consider the linear regression problems on synthetic data and the alternating minimization algorithm 1. We set the number of users $n = 50,000$, number of samples per user $m = 10$, data dimension $d = 50$ and rank $k = 2$. We sample $\mathbf{x}$, $\mathbf{U}$ and $\mathbf{v}$ from Gaussian distributions, and the field noise $\sigma_{\mathrm{F}}$ of target $y$ is set to be $0.01$. We normalize $\mathbf{U}$ to unit norm. We run Algorithm 1 with full batch, i.e., $T = 1$ and for multiple epochs. The privacy risk will accumulate over epochs, and we use the RDP sequential composition to account for that. We fix the clipping norm to be $10^{-4}$ and pick the optimal the number of epochs in $\{1, 2, 5, 10\}$.

In Figure 2, we fix $\delta = 10^{-6}$ and plot the population mean squared error (MSE) computed based on the groundtruth model for multiple $\varepsilon$. As a reference, the MSE for a purely random model is around 4. In addition to $(\varepsilon, \delta)$-differential privacy, we also report the value of Zero-Concentrated Differential Privacy (zCDP) [5], as it better captures the privacy properties, especially for Gaussian mechanism based algorithms. The definition can be found at Appendix A. In particular, privacy parameter $\varepsilon = \{1, 2, 5, 10\}$ corresponds to $0.009, 0.04, 0.23, 0.90$-zCDP, respectively. We observe that at $\varepsilon \approx 5$, the population MSE for Algorithm 3 is comparable to the non-private baseline. In contrast, the error for the single model baseline remains very high, even at high values of $\varepsilon$.

## 7  Conclusion

In this paper we studied the problem of personalized supervised learning with user-level differential privacy. Through our framework and Algorithm 1, we demonstrated that we can indeed learn accurate shared *linear* representation of the data, despite a limited number of samples-per-user and while preserving each user's privacy. Our error bounds and sample complexity bounds are nearly optimal in key parameters and are in fact, comparable to the best known bounds available for a much simpler linear regression problem.

This work leads to several interesting questions: (i) In our model, can we provide similar privacy/utility trade-offs for deep networks based embedding functions instead of a linear embedding function, (ii) Can we make a variant of the exponential mechanism algorithm computationally feasible?, and (iii) Empirically validate the privacy/utility trade-offs on real world data sets.

As more and more ML models are personalized for user tastes, ensuring privacy of individuals' data is paramount to a fair, responsible system. We provide a rigorous framework to design such solutions, which hopefully will motivate practitioners and researchers to make privacy as a first class citizen while designing their personalization based ML system.

## Acknowledgements and Funding Disclosure

We would like to thank Brendan McMahan, and Daniel Ramage for various helpful discussions during the course of this project. Adam Smith was supported in part by NSF award CCF-1763786 as well as a Sloan Foundation research award and a Google Faculty Research Award.

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
