## A  Additional Definitions

**Definition A.1** (zCDP [5])**.** *A randomized algorithm $\mathcal{A}$ is $\beta$-zCDP if for any pair of data sets $D$ and $D'$ that different in one record, we have $D_\alpha\left(\mathcal{A}(D)\|\mathcal{A}(D')\right) \leq \beta\alpha$ for all $\alpha > 1$, where $D_\alpha$ is the Rényi divergence of order $\alpha$.*

It is easy to see that $\beta$-zCDP is equivalent to $(\alpha, \beta\alpha)$-RDP for all order $\alpha$.

## B  Missing Proofs from Section 4

### B.1  Proof of Lemma 4.3

*Proof.* We will show that $\boldsymbol{W}_{\texttt{priv}}$ and $\boldsymbol{b}_{\texttt{priv}}$ in Algorithm 2 guarantee differential privacy. As the $\arg\min$ can be computed given the two quantities, it will guarantee differential privacy by sequential composition.

For any $j$, denote $\boldsymbol{A}_j = \sum_{i\in[m/4+1,m/2]} \boldsymbol{W}_{ij}\boldsymbol{W}_{ij}^\top$ and $\boldsymbol{b}_j = \sum_{i\in[m/4+1,m/2]} \widetilde{y}_{ij}\boldsymbol{W}_{ij}$. For any iteration $t$, let $\boldsymbol{A} = \sum_{j\in\mathcal{S}_t} \boldsymbol{A}_j$ and $\boldsymbol{b} = \sum_{j\in\mathcal{S}_t} \boldsymbol{b}_j$. Considering neighboring datasets $D$ and $D'$ such that user $j$'s data in $D$ is replaced by user $j^*$'s. If $j \notin \mathcal{S}_t$ in iteration $t$, $\boldsymbol{A}$ and $\boldsymbol{b}$ will be the same. Otherwise, $A$ would change by $\Delta\boldsymbol{A} = \boldsymbol{A}_{j^*} - \boldsymbol{A}_j$ and $\boldsymbol{b}$ by $\Delta\boldsymbol{b} = \boldsymbol{b}_{j^*} - \boldsymbol{b}_j$. We will bound the two quantities.

- For $\Delta\boldsymbol{A}$: According to the definitions, we have $\|\boldsymbol{W}_{ij}\|_2 \leq \eta$. Consider the Frobenius norm of matrix $\boldsymbol{W}_{ij}\boldsymbol{W}_{ij}^\top$. For any vector $x$, we have $\left\|\mathbf{x}\mathbf{x}^\top\right\|_F = \sqrt{\sum_{p,q} x_p^2 x_q^2} = \sqrt{\sum_p x_p^2 \sum_q x_q^2} = \|\mathbf{x}\|_2^2$. Therefore, we have $\left\|\boldsymbol{W}_{ij}\boldsymbol{W}_{ij}^\top\right\|_F = \|\boldsymbol{W}_{ij}\|_2^2 \leq \eta^2$, and thus $\|\boldsymbol{A}_j\|_F \leq m\eta^2/4$, and $\|\Delta\boldsymbol{A}\|_F \leq \|\boldsymbol{A}_j\|_F + \|\boldsymbol{A}_{j^*}\|_F \leq m\eta^2/2$.

- For $\Delta\boldsymbol{b}$: Again according to definition, we have $|\widetilde{y}_{ij}| \leq \zeta$ for any $j$. Thus $\|\boldsymbol{b}_j\|_2 \leq m\eta\zeta/4$ for any $j$, and $\|\Delta\boldsymbol{b}\|_2 \leq m\eta\zeta/2$.

Applying Gaussian mechanism, adding noise $\mathcal{N}(0, m^2\eta^2\zeta^2\Delta_{(\varepsilon,\delta)}^2/4)^{dk}$ to $\boldsymbol{b}$ guarantees $(\alpha, \alpha/(2\Delta_{(\varepsilon,\delta)}^2))$-RDP. As for $\boldsymbol{A}$, adding $\mathcal{N}(0, m^2\eta^4\Delta_{(\varepsilon,\delta)}^2/4)^{dk\times dk}$ to the vectorized version of $\boldsymbol{A}$ guarantees $(\alpha, \alpha/(2\Delta_{(\varepsilon,\delta)}^2))$-RDP. We can reshape the vectorized $\boldsymbol{A}$ to get the matrix version, which is a postprocessing step and does not affect the privacy guarantee. Notice that $\boldsymbol{A}$ is a symmetric matrix. We can thus copy its upper triangle to the lower, which is equivalent to adding a symmetric Gaussian matrix to $\boldsymbol{A}$ as stated in the algorithm.

By sequential composition, one run of Algorithm 2 guarantees $(\alpha, \alpha/\Delta_{(\varepsilon,\delta)}^2)$-RDP. Notice that Algorithm 1 calls Algorithm 2 for $T$ times on disjoint sets of users. So by parallel composition, Algorithm 1 guarantees $(\alpha, \alpha/\Delta_{(\varepsilon,\delta)}^2)$-RDP, which translates to $\left(\frac{\alpha}{\Delta_{(\varepsilon,\delta)}^2} + \frac{\log(1/\delta)}{\alpha-1}, \delta\right)$-DP for any $\varepsilon$, $\delta$ by standard conversion from RDP to approximate DP. Optimizing over $\alpha$, we get $\left(\frac{1}{\Delta_{(\varepsilon,\delta)}^2} + \frac{2\sqrt{\log(1/\delta)}}{\Delta_{(\varepsilon,\delta)}}, \delta\right)$-DP. Solving $\Delta_{(\varepsilon,\delta)}$ from $\frac{1}{\Delta_{(\varepsilon,\delta)}^2} + \frac{2\sqrt{\log(1/\delta)}}{\Delta_{(\varepsilon,\delta)}} \leq \varepsilon$, we have $\Delta_{(\varepsilon,\delta)} \geq \frac{\sqrt{\log(1/\delta)} + \sqrt{\log(1/\delta)+\varepsilon}}{\varepsilon}$. Therefore, if $\varepsilon \leq \log(1/\delta)$, it suffices to guarantee $(\varepsilon, \delta)$-DP by setting $\Delta_{(\varepsilon,\delta)} = \frac{\sqrt{8\log(1/\delta)}}{\varepsilon}$. $\qquad\square$

### B.2  Proof of Lemma 4.5

*Proof.* We will show that publishing $\boldsymbol{M}^{\texttt{Noisy}}$ guarantees differential privacy. As $\boldsymbol{W}_{ij}$'s and $\boldsymbol{M}^{\texttt{Noisy}}$ are all symmetric, for privacy analysis, it suffices to consider the upper triangles of them. Let $\mathsf{up}\,(X)$ denote the upper triangle of matrix $X$ flatten into a vector. Let $\boldsymbol{w}_{ij} = \mathsf{up}\,(\boldsymbol{W}_{ij})$, $\boldsymbol{w} = \sum_{i,j} \boldsymbol{w}_{ij}$, and $\widetilde{w} = \sum_{i,j} \boldsymbol{w}_{ij} + \mathsf{up}\left(\mathcal{N}_{\mathsf{sym}}\left(0, \Delta_{(\varepsilon,\delta)}^2 \zeta^4 m^2\right)^{d^2}\right)$. It is easy to see that $\boldsymbol{M}^{\texttt{Noisy}}$ can be formed by postprocessing $\widetilde{w}$. We will thus prove the privacy property of $\widetilde{w}$, which directly translate to the privacy guarantee of $\boldsymbol{M}^{\texttt{Noisy}}$.

Consider neighboring datasets $D$ and $D'$ such that user $j$'s data in $D$ is replaced by user $j^*$'s data in $D'$. Then the corresponding $\boldsymbol{w}$ would differ by $\sum_i \boldsymbol{w}_{ij^*} - \sum_i \boldsymbol{w}_{ij}$. We will analyze its $\ell_2$ norm. For

any $i$ and $j$, we have

$$\left\| \frac{\mathbf{x}_{(2i)j}\mathbf{x}_{(2i+1)j}^{\top}}{\left\|\mathbf{x}_{(2i)j}\right\|_{2} \cdot \left\|\mathbf{x}_{(2i+1)j}\right\|_{2}} \cdot \mathsf{clip}\left(y_{(2i)j};\zeta\right) \cdot \mathsf{clip}\left(y_{(2i+1)j};\zeta\right) \right\|_{F}$$

$$\leq \zeta^{2} \frac{\left\|\mathbf{x}_{(2i)j}\mathbf{x}_{(2i+1)j}^{\top}\right\|_{F}}{\left\|\mathbf{x}_{(2i)j}\right\|_{2} \cdot \left\|\mathbf{x}_{(2i+1)j}\right\|_{2}} = \zeta^{2}. \tag{3}$$

where $\|\cdot\|_F$ denotes the Frobenius norm. The inequality follows from the definition of the clipping operation, and the equality follows because for two vectors $a$, $b$, we have $\left\|ab^{\top}\right\|_2^2 = \sum_{p,q}(a_p b_q)^2 = \sum_p a_p^2 \cdot \sum_q b_q^2 = \|a\|_2^2 \|b\|_2^2$. Therefore, we have $\|\boldsymbol{w}_{ij}\|_2 \leq \zeta^2$ for any $i,j$, which implies $\|\sum_i \boldsymbol{w}_{ij^*} - \sum_i \boldsymbol{w}_{ij}\|_2 \leq \sum_i \|\boldsymbol{w}_{ij^*}\|_2 + \sum_i \|\boldsymbol{w}_{ij}\|_2 \leq m\zeta^2$ for any $j$, i.e., the $\ell_2$ sensitivity of $\boldsymbol{w}$ is $m\zeta^2$.

Using Gaussian mechanism, adding noise $\mathcal{N}(0, m^2\zeta^4\Delta_{(\varepsilon,\delta)}^2\mathbb{I})$ to $\boldsymbol{w}$ guarantees $(\alpha, \alpha/(2\Delta_{(\varepsilon,\delta)}^2))$-RDP for any order $\alpha \geq 1$, which translates to $\left(\frac{\alpha}{2\Delta_{(\varepsilon,\delta)}^2} + \frac{\log(1/\delta)}{\alpha-1}, \delta\right)$-DP for any $\varepsilon, \delta > 0$. Optimizing over $\alpha$, it translates to $\left(\frac{1}{2\Delta_{(\varepsilon,\delta)}^2} + \frac{\sqrt{2\log(1/\delta)}}{\Delta_{(\varepsilon,\delta)}}, \delta\right)$-DP. Solving $\frac{1}{2\Delta_{(\varepsilon,\delta)}^2} + \frac{\sqrt{2\log(1/\delta)}}{\Delta_{(\varepsilon,\delta)}} \leq \varepsilon$, we get $\Delta_{(\varepsilon,\delta)} \geq \frac{\sqrt{\log(1/\delta)}+\sqrt{\log(1/\delta)+\varepsilon}}{\sqrt{2\varepsilon}}$. Therefore, if $\varepsilon \leq \log(1/\delta)$, it suffices to guarantee $(\varepsilon, \delta)$-DP by setting $\Delta_{(\varepsilon,\delta)} = \frac{\sqrt{8\log(1/\delta)}}{\varepsilon}$. $\qquad\square$

### B.3   Proof of Lemma 4.6

*Proof.* Let $\boldsymbol{M} = \frac{2}{nm} \sum_{i\in[m/2],j\in[n]} \boldsymbol{W}_{ij}$ and $\boldsymbol{U}^{\texttt{non-priv}}$ be the matrix with the top-$k$ eigenvectors of $\boldsymbol{M}$ as columns. Let $\Pi^{\texttt{priv}} = \boldsymbol{U}^{\texttt{priv}}\left(\boldsymbol{U}^{\texttt{priv}}\right)^{\top}$ and $\Pi^* = \boldsymbol{U}^*(\boldsymbol{U}^*)^{\top}$. Notice that $\left\|\Pi^* - \Pi^{\texttt{priv}}\right\|_2 \leq \left\|\Pi^* - \Pi^{\texttt{non-priv}}\right\|_2 + \left\|\Pi^{\texttt{non-priv}} - \Pi^{\texttt{priv}}\right\|_2$. We bound the first term via Lemma B.1 below. In order to bound the second term, first notice that the $k$-th eigenvalue of $\boldsymbol{M}$ (in Algorithm 3) (denoted by $\widehat{\lambda}_k$) is lower bounded as follows. This follows with high probability from (18) by choosing appropriate $\beta$ in Lemma B.1, polynomial in $n^{-1}$.

$$\widehat{\lambda}_k \geq \frac{\lambda_k}{d} - O\left(\sqrt{\frac{\mu^4 k^2 \lambda_k \log(dn)}{dnm}}\right) = \Omega\left(\frac{\lambda_k}{d}\right) \tag{4}$$

Now, we can use [17, Theorem 7] to directly bound $\left\|\Pi^{\texttt{non-priv}} - \Pi^{\texttt{priv}}\right\|_F = O\left(\frac{\Delta_{(\varepsilon,\delta)}d\sqrt{dk\log(dn)}}{n\cdot\lambda_k}\right)$, and correspondingly $\left\|\Pi^{\texttt{non-priv}} - \Pi^{\texttt{priv}}\right\|_2 = O\left(\frac{\zeta^2\Delta_{(\varepsilon,\delta)}d\sqrt{d\log(dn)}}{n\cdot\lambda_k}\right)$. Setting $\zeta$ as in the lemma statement, and observing rotation invariant property of the norms, completes the proof. $\qquad\square$

**Lemma B.1** (Non-private subspace closeness). *Let* $\Pi^{non\text{-}priv} = \boldsymbol{U}^{non\text{-}priv}\left(\boldsymbol{U}^{non\text{-}priv}\right)^{\top}$, *and* $\Pi^* = \boldsymbol{U}^*(\boldsymbol{U}^*)^{\top}$. *Following the assumption in Lemma 4.6, we have the following for Algorithm 3 (Algorithm $\mathcal{A}_{\mathsf{Priv\text{-}init}}$) w.p. at least $1 - \beta$ (over the randomness of data generation and the algorithm):*

$$\left\|\Pi^* - \Pi^{non\text{-}priv}\right\|_2 = \widetilde{O}\left(\sqrt{\frac{d\zeta^4\log(d/\beta)}{\lambda_k^2 nm}}\right).$$

*Proof.* By Gaussian concentration we have w.p. at least $1 - \beta/2$, $\forall i \in [m], j \in [n]$, $|\langle\mathbf{x}_{ij}, \boldsymbol{U}^*\cdot\boldsymbol{v}_j^*\rangle| \leq \mu\sqrt{k\lambda_k} \cdot \sqrt{2\ln(4nm/\beta)}$ and $|\boldsymbol{z}_{ij}| \leq \sigma_{\mathsf{F}}\sqrt{2\ln(4nm/\beta)}$. Hence, if we set the clipping threshold for the response $y_{ij}$ to be $\zeta = \left(\mu\sqrt{k\lambda_k} + \sigma_{\mathsf{F}}\right)\sqrt{2\ln(4nm/\beta)}$, then w.p. at least $1 - \beta/2$, clipping will not have any impact on the analysis. Call this event $\mathcal{A}$. We will perform the linear-algebra analysis

below without conditioning on this event, but our application of matrix Bernstein [50, Theorem 1.4] will rely on this bound.

We first note that for a Gaussian random vector $\mathbf{x}$, we have

$$\mathbb{E}\left[\frac{\mathbf{x}}{\|\mathbf{x}\|_2}\mathbf{x}^\top\right] = \mathbb{E}\left[\frac{\mathbf{x}\mathbf{x}^\top}{\mathbf{x}^\top\mathbf{x}}\|\mathbf{x}\|_2\right] = \frac{\mathbb{I}}{d}\cdot\mathbb{E}\left[\|\mathbf{x}\|_2\right] = \frac{\Gamma\left(\frac{d+1}{2}\right)}{d\sqrt{2}\Gamma\left(\frac{d}{2}\right)}\mathbb{I} \simeq \frac{1}{\sqrt{d}}\mathbb{I} \tag{5}$$

This can be seen by first noting that the magnitude of a random Gaussian vector is independent of its direction (i.e., the Gaussian measure with identity covariance is a product measure in spherical coordinates, trivial from the fact that it is spherically symmetric), then explicitly evaluating the expected normalized outer product $\frac{\mathbf{x}\mathbf{x}^\top}{\mathbf{x}\cdot\mathbf{x}}$. Term-by-term, this evaluation reduces to $\mathbb{E}\left[\frac{\mathbf{x}[i]\mathbf{x}[j]}{\sum_{i=1}^d\mathbf{x}[i]^2}\right]$. Symmetry implies this expectation is 0 for $i\neq j$ and $\frac{1}{d}$ for $i=j$. Finally we apply a well-known formula for the expected Euclidean norm of a Gaussian random vector [45]. We now have (6) and (7) (as a measure of bias and variance) for any $i\in[m/2], j\in[n]$. Here, $\|\boldsymbol{W}_{ij}\|_2$ is the operator norm of $\boldsymbol{W}_{ij}$.

$$\mathbb{E}\left[\boldsymbol{W}_{ij}\right] = \mathbb{E}\left[\frac{\mathbf{x}_{(2i)j}}{\|\mathbf{x}_{(2i)j}\|_2}\mathbf{x}_{(2i)j}^\top\left(\boldsymbol{U}^*\boldsymbol{v}_j^*\left(\boldsymbol{v}_j^*\right)^\top\left(\boldsymbol{U}^*\right)^\top\right)\cdot\frac{\mathbf{x}_{(2i+1)j}}{\|\mathbf{x}_{(2i+1)j}\|_2}\mathbf{x}_{(2i+1)j}^\top\right] \simeq \frac{1}{d}\boldsymbol{U}^*\left(\boldsymbol{v}_j^*\left(\boldsymbol{v}_j^*\right)^\top\right)\left(\boldsymbol{U}^*\right)^\top \tag{6}$$

$$\|\boldsymbol{W}_{ij}\|_2 \leq \zeta^2 \tag{7}$$

Therefore, by (6) we have the following. Here, $\boldsymbol{V}^* = [\boldsymbol{v}_1^*|\cdots|\boldsymbol{v}_n^*]$.

$$\boldsymbol{B} = \frac{4}{nm}\sum_{i\in[m/4],j\in[n]}\mathbb{E}\left[\boldsymbol{W}_{ij}\right] \simeq \boldsymbol{U}^*\left(\frac{1}{dn}\sum_{j=1}^n\boldsymbol{v}_j^*\left(\boldsymbol{v}_j^*\right)^\top\right)\left(\boldsymbol{U}^*\right)^\top = \frac{1}{dn}\boldsymbol{U}^*\left(\boldsymbol{V}^*\left(\boldsymbol{V}^*\right)^\top\right)\left(\boldsymbol{U}^*\right)^\top \tag{8}$$

We will now bound $\left\|\frac{4}{nm}\sum_{i\in[m/4],j\in[n]}\boldsymbol{W}_{ij} - \boldsymbol{B}\right\|_2$ using Matrix Bernstein's inequality [49, Theorem 1.4]. Let $\boldsymbol{A}_{ij} = \boldsymbol{W}_{ij} - \frac{1}{d}\cdot\boldsymbol{U}^*\left(\boldsymbol{v}_j^*\left(\boldsymbol{v}_j^*\right)^\top\right)\left(\boldsymbol{U}^*\right)^\top$. Clearly, $\mathbb{E}\left[\boldsymbol{A}_{ij}\right]=0$, and $\|\boldsymbol{A}_{ij}\cdot 1_{\mathcal{A}}\|_2\leq\zeta^2+\frac{C^2}{d}$.

Now, in the following we bound $\left\|\sum_{i\in[m/4],j\in[n]}\mathbb{E}\left[\boldsymbol{A}_{ij}^2\right]\right\|_2$. Let $\Pi_j^*$ be the projector onto the eigenspace of $\boldsymbol{U}^*\boldsymbol{v}_j^*\left(\boldsymbol{v}_j^*\right)^\top\left(\boldsymbol{U}^*\right)^\top$. We have the following in (9).

$$\sum_{i\in[m/4],j\in[n]}\mathbb{E}\left[\boldsymbol{A}_{ij}^2\right] = \sum_{i\in[m/4],j\in[n]}\mathbb{E}\left[\boldsymbol{W}_{ij}^2\right] - \frac{m}{4d^2}\sum_{j\in[n]}\boldsymbol{U}^*\boldsymbol{v}_j^*\left(\boldsymbol{v}_j^*\right)^\top\left(\boldsymbol{U}^*\right)^\top\boldsymbol{U}^*\boldsymbol{v}_j^*\left(\boldsymbol{v}_j^*\right)^\top\left(\boldsymbol{U}^*\right)^\top$$

$$= \sum_{i\in[m/4],j\in[n]}\mathbb{E}\left[\boldsymbol{W}_{ij}^2\right] - \frac{m}{4d^2}\sum_{j\in[n]}\left\|\boldsymbol{U}^*\boldsymbol{v}_j^*\right\|_2^4\cdot\Pi_j^* \tag{9}$$

We now bound $\mathbb{E}\left[\boldsymbol{W}_{ij}^2\right]$ the first term in (9). We have the following.

$$\mathbb{E}\left[\boldsymbol{W}_{ij}^2\right] = \mathbb{E}\left[\frac{\mathbf{x}_{(2i)j}\mathbf{x}_{(2i)j}^\top}{\|\mathbf{x}_{(2i)j}\|_2}\boldsymbol{U}^*\left(\boldsymbol{v}_j^*\left(\boldsymbol{v}_j^*\right)^\top\right)\left(\boldsymbol{U}^*\right)^\top\frac{\mathbf{x}_{(2i+1)j}\mathbf{x}_{(2i+1)j}^\top}{\|\mathbf{x}_{(2i+1)j}\|_2}\frac{\mathbf{x}_{(2i+1)j}\mathbf{x}_{(2i+1)j}^\top}{\|\mathbf{x}_{(2i+1)j}\|_2}\boldsymbol{U}^*\left(\boldsymbol{v}_j^*\left(\boldsymbol{v}_j^*\right)^\top\right)\left(\boldsymbol{U}^*\right)^\top\frac{\mathbf{x}_{(2i)j}\mathbf{x}_{(2i)j}^\top}{\|\mathbf{x}_{(2i)j}\|_2}\right]$$

$$= \mathbb{E}\left[\frac{1}{\|\mathbf{x}_{(2i)j}\|_2^2}\mathbf{x}_{(2i)j}\mathbf{x}_{(2i)j}^\top\cdot\boldsymbol{U}^*\left(\boldsymbol{v}_j^*\left(\boldsymbol{v}_j^*\right)^\top\right)\left(\boldsymbol{U}^*\right)^\top\mathbf{x}_{(2i+1)j}\mathbf{x}_{(2i+1)j}^\top\boldsymbol{U}^*\left(\boldsymbol{v}_j^*\left(\boldsymbol{v}_j^*\right)^\top\right)\left(\boldsymbol{U}^*\right)^\top\mathbf{x}_{(2i)j}\mathbf{x}_{(2i)j}^\top\right]$$

$$= \mathbb{E}\left[\frac{1}{\|\mathbf{x}_{(2i)j}\|_2^2}\mathbf{x}_{(2i)j}\mathbf{x}_{(2i)j}^\top\cdot\boldsymbol{U}^*\left(\boldsymbol{v}_j^*\left(\boldsymbol{v}_j^*\right)^\top\right)\left(\boldsymbol{U}^*\right)^\top\boldsymbol{U}^*\left(\boldsymbol{v}_j^*\left(\boldsymbol{v}_j^*\right)^\top\right)\left(\boldsymbol{U}^*\right)^\top\mathbf{x}_{(2i)j}\mathbf{x}_{(2i)j}^\top\right] \tag{10}$$

In the last equality, we have used independence to evaluate the outer product in the middle of the expression. This operation can be viewed as evaluating a chain of conditional expectations: $\mathbb{E}\left[\boldsymbol{ABA}\right] = \mathbb{E}\left[\mathbb{E}\left[\boldsymbol{ABA}|\boldsymbol{A}\right]\right] = \mathbb{E}\left[\boldsymbol{A} \cdot \mathbb{E}\left[\boldsymbol{B}|\boldsymbol{A}\right] \cdot \boldsymbol{A}\right] = \mathbb{E}\left[\boldsymbol{A} \cdot \mathbb{E}\left[\boldsymbol{B}\right] \cdot \boldsymbol{A}\right]$. Separating the norm of $\boldsymbol{U}^*\boldsymbol{v}_j^*(\boldsymbol{U}^*\boldsymbol{v}_j^*)^\top$ from projection onto its range, we see

$$
\begin{aligned}
\mathbb{E}\left[\boldsymbol{W}_{ij}^2\right] &= \mathbb{E}\left[\frac{\left\|\boldsymbol{U}^*\boldsymbol{v}_j^*\right\|_2^4}{\left\|\mathbf{x}_{(2i)j}\right\|_2^2} \mathbf{x}_{(2i)j}\mathbf{x}_{(2i)j}^\top \cdot \Pi_j^* \cdot \mathbf{x}_{(2i)j}\mathbf{x}_{(2i)j}^\top\right] \\
&= \mathbb{E}\left[\frac{\left\|\boldsymbol{U}^*\boldsymbol{v}_j^*\right\|_2^4}{\left\|\mathbf{x}_{(2i)j}\right\|_2^2} \mathbf{x}_{(2i)j}\mathbf{x}_{(2i)j}^\top \cdot \left(\Pi_j^*\right)^\top \cdot \Pi_j^* \cdot \mathbf{x}_{(2i)j}\mathbf{x}_{(2i)j}^\top\right] \\
&= \left\|\boldsymbol{U}^*\boldsymbol{v}_j^*\right\|_2^4 \cdot \mathbb{E}\left[\left\|\Pi_j^*\mathbf{x}_{(2i)j}\right\|_2^2 \cdot \frac{\mathbf{x}_{(2i)j}\mathbf{x}_{(2i)j}^\top}{\left\|\mathbf{x}_{(2i)j}\right\|_2^2}\right] \quad (11)
\end{aligned}
$$

To estimate the expectation on the right, we let $\boldsymbol{a} = \Pi_j^*\mathbf{x}_{(2i)j}$ and $\boldsymbol{b} = (\mathbb{I} - \Pi_j^*)\mathbf{x}_{(2i)j}$, and note that $\boldsymbol{a}$ and $\boldsymbol{b}$ are independent. So we are interested in evaluating

$$
\mathbb{E}\left[\left\|\boldsymbol{a}\right\|_2^2 \frac{(\boldsymbol{a}+\boldsymbol{b})(\boldsymbol{a}+\boldsymbol{b})^\top}{\left\|\boldsymbol{a}\right\|_2^2 + \left\|\boldsymbol{b}\right\|_2^2}\right] = \mathbb{E}\left[\frac{\left\|\boldsymbol{a}\right\|_2^2}{\left\|\boldsymbol{a}\right\|_2^2 + \left\|\boldsymbol{b}\right\|_2^2}(\boldsymbol{a}\boldsymbol{a}^\top + \boldsymbol{b}\boldsymbol{b}^\top)\right] + \mathbb{E}\left[\frac{\left\|\boldsymbol{a}\right\|_2^2}{\left\|\boldsymbol{a}\right\|_2^2 + \left\|\boldsymbol{b}\right\|_2^2}(\boldsymbol{a}\boldsymbol{b}^\top + \boldsymbol{b}\boldsymbol{a}^\top)\right]
$$
(12)

The second expectation is 0, as can be noted by symmetry. That is, conditioning on $\boldsymbol{b}$ and $\left\|\boldsymbol{a}\right\|_2$ yields the integral of a spherically symmetric random variable. We can then bound:

$$
\begin{aligned}
\mathbb{E}\left[\left\|\boldsymbol{a}\right\|_2^2 \frac{(\boldsymbol{a}+\boldsymbol{b})(\boldsymbol{a}+\boldsymbol{b})^\top}{\left\|\boldsymbol{a}\right\|_2^2 + \left\|\boldsymbol{b}\right\|_2^2}\right] &\preccurlyeq \mathbb{E}\left[\frac{\left\|\boldsymbol{a}\right\|_2^2}{\left\|\boldsymbol{b}\right\|_2^2}\boldsymbol{a}\boldsymbol{a}^\top\right] + \mathbb{E}\left[\left\|\boldsymbol{a}\right\|_2^2\right]\mathbb{E}\left[\frac{\boldsymbol{b}\boldsymbol{b}^\top}{\left\|\boldsymbol{b}\right\|_2^2}\right] \\
&= \mathbb{E}\left[\frac{1}{\left\|\boldsymbol{b}\right\|_2^2}\right]\mathbb{E}\left[\left\|\boldsymbol{a}\right\|_2^4\right]\Pi_j^* + \eta\left(\mathbb{I} - \Pi_j^*\right) \quad (13)
\end{aligned}
$$

for some $\eta > 0$. $\mathbb{E}\left[\frac{1}{\left\|\boldsymbol{b}\right\|_2^2}\right] = O\left(\frac{1}{d}\right)$ and $\mathbb{E}\left[\left\|\boldsymbol{a}\right\|_2^4\right] = O(1)$, so the first term is on the order of $\frac{1}{d} \cdot \Pi_j^*$. We evaluate $\eta$ by cyclically permuting the trace:

$$
\eta(d-1) = \mathbf{tr}\left(\eta\left(\mathbb{I} - \Pi_j^*\right)\right) = \mathbf{tr}\left(\mathbb{E}\left[\frac{\boldsymbol{b}\boldsymbol{b}^\top}{\left\|\boldsymbol{b}\right\|_2^2}\right]\right) = \mathbb{E}\left[\mathbf{tr}\left(\frac{\boldsymbol{b}\boldsymbol{b}^\top}{\left\|\boldsymbol{b}\right\|_2^2}\right)\right] = \mathbb{E}\left[\mathbf{tr}\left(\frac{\boldsymbol{b}^\top\boldsymbol{b}}{\left\|\boldsymbol{b}\right\|_2^2}\right)\right] = 1
$$
(14)

so that $\eta = \frac{1}{d-1} = O\left(\frac{1}{d}\right)$.

Putting together (13) and (14) with (11), we see

$$
\mathbb{E}\left[\boldsymbol{W}_{ij}^2\right] \preccurlyeq O\left(\frac{\left\|\boldsymbol{U}^*\boldsymbol{v}_j^*\right\|_2^4}{d}\right) \cdot \mathbb{I} \quad (15)
$$

From (9) and (15) we have the following.

$$
\left\|\sum_{i\in[m/2],j\in[n]}\mathbb{E}\left[\boldsymbol{A}_{ij}^2\right]\right\|_2 = O\left(\frac{m}{d}\sum_{j\in[n]}\left\|\boldsymbol{U}^*\boldsymbol{v}_j^*\right\|_2^4\right) = O\left(\frac{mn\mu^4 k^2 \lambda_k^2}{d}\right) \quad (16)
$$

Therefore we may apply Matrix Bernstein's inequality [50, Theorem 1.4] by restricting nonzero values to the previously defined event $\mathcal{A}$ where clipping plays no role, ensuring the pointwise bound

$\|\boldsymbol{A}_{ij} \cdot 1_{\mathcal{A}}\|_2 \leq \zeta^2 + \frac{\mu^2 k \lambda_k}{d}$. Notice that this restriction can only strengthen the bound (16). So we have the following.

$$\mathbf{Pr}\left[\left\|\frac{4}{nm}\sum_{i\in[m/4],j\in[n]}\boldsymbol{A}_{ij}\cdot 1_{\mathcal{A}}\right\|_2 \geq \frac{4t}{nm}\right] \leq d\cdot\exp\left(-\frac{t^2/2}{O\left(\frac{nm\mu^4 k^2 \lambda_k^2}{d}\right)+\left(\zeta^2+\frac{C^2}{d}\right)\cdot\frac{t}{3}}\right) \leq \frac{\beta}{2}$$
$$(17)$$

Setting $t = \sqrt{\log(d/\beta)} \cdot \Omega\left(\max\left\{\sqrt{\frac{nm\mu^4 k^2 \lambda_k^2}{d}}, \left(\zeta^2 + \frac{\mu^2 k\lambda_k}{d}\right)\sqrt{\log(d/\beta)}\right\}\right)$ in (17) suffices, by setting up and solving the associated quadratic. Therefore, since $\mathbb{P}\left[\mathcal{A}^c\right] \leq \frac{\beta}{2}$, w.p. at least $1 - \beta$ we have:

$$\left\|\frac{4}{nm}\sum_{i\in[m/4],j\in[n]}\boldsymbol{A}_{ij}\right\|_2 \leq \sqrt{\log(d/\beta)}\cdot O\left(\max\left\{\frac{\mu^2 k\lambda_k}{\sqrt{dnm}}, \frac{\left(\zeta^2+\mu^2 k\lambda_k/d\right)\sqrt{\log(d/\beta)}}{nm}\right\}\right) = O\left(\sqrt{\frac{\zeta^4\cdot\log(d/\beta)}{dnm}}\right)$$
$$(18)$$

The last equality in (18) follows from the assumption $mn = \Omega\left(d\left(\zeta^2 + \frac{\mu^2 k\lambda_k}{d}\right)^2 \cdot \log(d/\beta)/(\mu^2 k\lambda_k)^2\right)$. With (18) in hand, we now use the Davis-Kahn Sin $\Theta$-theorem [12] from matrix perturbation theory to bound $\left\|\Pi^{\texttt{non-priv}} - \Pi^*\right\|_2$. We use the following variant in Lemma B.2.

**Lemma B.2** (Sin $\Theta$-Theorem [12]). *Let $\boldsymbol{G}$ and $\boldsymbol{H}$ be two PSD matrices. Let $\Pi_{\boldsymbol{G}}^{(i)}$ be the projector onto the top-$i$ eigenvectors of $\boldsymbol{G}$, and let $\mathsf{eig}^{(i)}(\boldsymbol{G})$ be the $i$-th largest eigenvalue of $\boldsymbol{G}$. Define these quantities correspondingly for $\boldsymbol{H}$. Then, the following is true.*

$$\left(\mathsf{eig}^{(i)}(\boldsymbol{G}) - \mathsf{eig}^{(j+1)}(\boldsymbol{G})\right) \cdot \left(\left(\mathbb{I} - \Pi_{\boldsymbol{H}}^{(j)}\right)\Pi_{\boldsymbol{G}}^{(i)}\right) \leq \|\boldsymbol{G} - \boldsymbol{H}\|_2$$

Let $\boldsymbol{G} = \frac{1}{dn}\boldsymbol{U}^*\left(\boldsymbol{V}^*\left(\boldsymbol{V}^*\right)^\top\right)\left(\boldsymbol{U}^*\right)^\top$ and $\boldsymbol{H} = \frac{4}{nm}\sum_{i\in[m/4],j\in[n]}\boldsymbol{W}_{ij}$. Note that both $\boldsymbol{G}$ and $\boldsymbol{H}$ are PSD matrices. Furthermore, from (18) we have $\|\boldsymbol{G} - \boldsymbol{H}\|_2 = O\left(\sqrt{\frac{\zeta^4\cdot\log(d/\beta)}{dnm}}\right)$ w.p. $\geq 1 - \beta$. Recall that $\Pi^{\texttt{non-priv}}$ is the projector onto the rank-$k$ approximation of $\boldsymbol{H}$. Following the notation of Lemma B.2, and by assumption $\sqrt{nm} = \Omega\left(\sqrt{d\zeta^4\log(d/\beta)}/\lambda_k\right)$, we have $\mathsf{eig}^{(k)}(\boldsymbol{G}) = \frac{\lambda_k}{d}$, $\mathsf{eig}^{(k)}\left(\Pi^{\texttt{non-priv}}\right) \in \left[\frac{\mathsf{eig}^{(k)}(\boldsymbol{G})}{2}, 2\cdot\mathsf{eig}^{(k)}(\boldsymbol{G})\right]$, and $\mathsf{eig}^{(k+1)}\left(\Pi^{\texttt{non-priv}}\right) \leq \frac{\mathsf{eig}^{(k)}(\boldsymbol{G})}{2}$. Here, $\lambda_k$ is the $k$-th eigenvalue of $\boldsymbol{U}^*\left(\frac{1}{n}\boldsymbol{V}^*\left(\boldsymbol{V}^*\right)^\top\right)\left(\boldsymbol{U}^*\right)^\top$, which equals the $k$-th eigenvalue of $\frac{1}{n}\boldsymbol{V}^*\left(\boldsymbol{V}^*\right)^\top$. Also, notice that the projector onto $\boldsymbol{G}$ equals $\Pi^*$ as long as $\lambda_k > 0$, which is true by assumption.

Therefore, from Lemma B.2 we have the following w.p. at least $1 - \beta$.

$$\left\|\left(\mathbb{I} - \Pi^*\right)\Pi^{\texttt{non-priv}}\right\|_2 = O\left(\frac{\sqrt{\frac{\zeta^4\cdot\log(d/\beta)}{dnm}}}{\mathsf{eig}^{(k)}(\boldsymbol{G})}\right) \tag{19}$$

$$\left\|\left(\mathbb{I} - \Pi^{\texttt{non-priv}}\right)\Pi^*\right\|_2 = O\left(\frac{\sqrt{\frac{\zeta^4\cdot\log(d/\beta)}{dnm}}}{\mathsf{eig}^{(k)}(\boldsymbol{G})}\right) \tag{20}$$

Furthermore, notice that $\left\|\Pi^* - \Pi^{\texttt{non-priv}}\right\|_2 \leq \left\|\left(\mathbb{I} - \Pi^*\right)\Pi^{\texttt{non-priv}}\right\|_2 + \left\|\left(\mathbb{I} - \Pi^{\texttt{non-priv}}\right)\Pi^*\right\|_2$. Plugging in the value of $\mathsf{eig}^{(k)}(\boldsymbol{G})$ in (19) and (20) completes the proof. $\qquad\square$

## B.4 Proof of Theorem 4.2

*Proof.* Let $b = \langle \boldsymbol{a}, \boldsymbol{U}^* \boldsymbol{v}^* \rangle + w$, where $\boldsymbol{a} \sim \mathcal{N}(0,1)^d$, $w \sim \mathcal{N}(0, \sigma_{\mathrm{F}}^2)$, $\boldsymbol{U}^* \in \mathbb{R}^{d \times k}$ is a matrix with orthonormal columns, and $\boldsymbol{v}^* \in \mathbb{R}^k$. Consider the loss function $\mathcal{L}(\boldsymbol{U}, \boldsymbol{v}) = \mathbb{E}_{\boldsymbol{a}, w}\left[(b - \langle \boldsymbol{a}, \boldsymbol{U} \boldsymbol{v} \rangle)^2\right]$, where $\boldsymbol{U} \in \mathbb{R}^{d \times k}$ is a matrix with orthonormal columns and $\boldsymbol{v} \in \mathbb{R}^k$. We have,

$$
\begin{aligned}
\mathcal{L}(\boldsymbol{U}, \boldsymbol{v}) &= \mathbb{E}\left[\left(\boldsymbol{a}^\top \left(\boldsymbol{U}^* \boldsymbol{v}^* - \boldsymbol{U} \boldsymbol{v}\right) + w\right)^2\right] \\
&= \left(\boldsymbol{U}^* \boldsymbol{v}^* - \boldsymbol{U} \boldsymbol{v}\right)^\top \mathbb{E}\left[\boldsymbol{a} \boldsymbol{a}^\top\right] \left(\boldsymbol{U}^* \boldsymbol{v}^* - \boldsymbol{U} \boldsymbol{v}\right) + \sigma_{\mathrm{F}}^2 \\
&= \left\|\boldsymbol{U}^* \boldsymbol{v}^* - \boldsymbol{U} \boldsymbol{v}\right\|_2^2 + \sigma_{\mathrm{F}}^2.
\end{aligned} \tag{21}
$$

We consider $\widehat{\boldsymbol{v}} = \arg\min_{\boldsymbol{v}} \left\|\boldsymbol{y} - \boldsymbol{X}^\top \widehat{\boldsymbol{U}} \boldsymbol{v}\right\|_2^2 = \left(\widehat{\boldsymbol{U}}^\top \boldsymbol{X} \boldsymbol{X}^\top \widehat{\boldsymbol{U}}\right)^{-1} \widehat{\boldsymbol{U}}^\top \boldsymbol{X} \boldsymbol{y}$, where $\widehat{\boldsymbol{U}} \in \mathbb{R}^{d \times k}$ is some matrix with orthonormal columns, $\boldsymbol{X} \sim \mathcal{N}(0,1)^{d \times m}$ and $\boldsymbol{y} = \boldsymbol{X}^\top \boldsymbol{U}^* \boldsymbol{v}^* + \boldsymbol{w}$ (with $\boldsymbol{w} \sim \mathcal{N}(0, \sigma_{\mathrm{F}}^2)^m$). Notice that the inverse exists w.p. at least $1 - \frac{1}{m^{10}}$ as long as $m = \Omega(k)$.

In the following, we will bound $\mathcal{L}(\widehat{\boldsymbol{U}}, \widehat{\boldsymbol{v}})$. To do so, we will first bound $\left\|\boldsymbol{U}^* \boldsymbol{v}^* - \widehat{\boldsymbol{U}} \boldsymbol{v}\right\|_2^2$ in (21). Assume, $\widehat{\Pi} = \widehat{\boldsymbol{U}} \widehat{\boldsymbol{U}}^\top$, $\Pi^* = \boldsymbol{U}^* (\boldsymbol{U}^*)^\top$, $\Delta = \widehat{\Pi} - \Pi^*$, and $\|\Delta\|_2 \leq \Gamma$. We have,

$$
\begin{aligned}
\mathbb{E}\left[\left\|\boldsymbol{U}^* \boldsymbol{v}^* - \widehat{\boldsymbol{U}} \widehat{\boldsymbol{v}}\right\|_2^2\right] &= \mathbb{E}\left[\left\|\widehat{\boldsymbol{U}}\left(\widehat{\boldsymbol{U}}^\top \boldsymbol{X} \boldsymbol{X}^\top \widehat{\boldsymbol{U}}\right)^{-1} \widehat{\boldsymbol{U}}^\top \boldsymbol{X} \boldsymbol{y} - \boldsymbol{U}^* \boldsymbol{v}^*\right\|_2^2\right] \\
&= \mathbb{E}\left[\left\|\widehat{\boldsymbol{U}}\left(\widehat{\boldsymbol{U}}^\top \boldsymbol{X} \boldsymbol{X}^\top \widehat{\boldsymbol{U}}\right)^{-1} \widehat{\boldsymbol{U}}^\top \boldsymbol{X} \boldsymbol{X}^\top \boldsymbol{U}^* \boldsymbol{v}^* - \boldsymbol{U}^* \boldsymbol{v}^* + \widehat{\boldsymbol{U}}\left(\widehat{\boldsymbol{U}}^\top \boldsymbol{X} \boldsymbol{X}^\top \widehat{\boldsymbol{U}}\right)^{-1} \widehat{\boldsymbol{U}}^\top \boldsymbol{X} \boldsymbol{w}\right\|_2^2\right] \\
&= \mathbb{E}\left[\left\|\widehat{\boldsymbol{U}}\left(\widehat{\boldsymbol{U}}^\top \boldsymbol{X} \boldsymbol{X}^\top \widehat{\boldsymbol{U}}\right)^{-1} \widehat{\boldsymbol{U}}^\top \boldsymbol{X} \boldsymbol{X}^\top \boldsymbol{U}^* \boldsymbol{v}^* - \boldsymbol{U}^* \boldsymbol{v}^*\right\|_2^2\right] + \frac{k}{m} \sigma_{\mathrm{F}}^2 \\
&= \mathbb{E}\left[\left\|\widehat{\boldsymbol{U}}\left(\widehat{\boldsymbol{U}}^\top \boldsymbol{X} \boldsymbol{X}^\top \widehat{\boldsymbol{U}}\right)^{-1} \widehat{\boldsymbol{U}}^\top \boldsymbol{X} \boldsymbol{X}^\top \left(\widehat{\boldsymbol{U}} \widehat{\boldsymbol{U}}^\top \cdot \boldsymbol{U}^* \boldsymbol{v}^* + (\mathbb{I} - \widehat{\boldsymbol{U}} \widehat{\boldsymbol{U}}^\top) \boldsymbol{U}^* \boldsymbol{v}^*\right) - \boldsymbol{U}^* \boldsymbol{v}^*\right\|_2^2\right] + \frac{k}{m} \sigma_{\mathrm{F}}^2 \\
&= \mathbb{E}\left[\left\|\widehat{\boldsymbol{U}}\left(\widehat{\boldsymbol{U}}^\top \boldsymbol{X} \boldsymbol{X}^\top \widehat{\boldsymbol{U}}\right)^{-1} \widehat{\boldsymbol{U}}^\top \boldsymbol{X} \boldsymbol{X}^\top \widehat{\boldsymbol{U}} \widehat{\boldsymbol{U}}^\top \boldsymbol{U}^* \boldsymbol{v}^* - \boldsymbol{U}^* \boldsymbol{v}^*\right\|_2^2\right] + \frac{k}{m} \sigma_{\mathrm{F}}^2 \\
&= \left\|\widehat{\boldsymbol{U}} \widehat{\boldsymbol{U}}^\top \boldsymbol{U}^* \boldsymbol{v}^* - \boldsymbol{U}^* \boldsymbol{v}^*\right\|_2^2 + \frac{k}{m} \sigma_{\mathrm{F}}^2 \\
&= \left\|(\Pi^* + \Delta) \boldsymbol{U}^* \boldsymbol{v}^* - \boldsymbol{U}^* \boldsymbol{v}^*\right\|_2^2 + \frac{k}{m} \sigma_{\mathrm{F}}^2 \\
&= \left\|\Delta \boldsymbol{U}^* \boldsymbol{v}^*\right\|_2^2 + \frac{k}{m} \sigma_{\mathrm{F}}^2 \\
&\leq \Gamma^2 \left\|\boldsymbol{U}^* \boldsymbol{v}^*\right\|_2^2 + \frac{k}{m} \sigma_{\mathrm{F}}^2
\end{aligned} \tag{22}
$$

Therefore, by (22) and (21), we have the following.

$$
\mathbb{E}\left[\mathcal{L}(\widehat{\boldsymbol{U}}, \widehat{\boldsymbol{v}})\right] \leq \Gamma^2 \left\|\boldsymbol{U}^* \boldsymbol{v}^*\right\|_2^2 + \left(\frac{k}{m} + 1\right) \sigma_{\mathrm{F}}^2 \tag{23}
$$

Let $\Pi^{\mathtt{priv}} = \boldsymbol{U}^{\mathtt{priv}} \left(\boldsymbol{U}^{\mathtt{priv}}\right)^\top$. (23) immediately implies,

$$
\mathsf{Risk}_{\mathtt{Pop}}\left(\left(\boldsymbol{U}^{\mathtt{priv}}, \boldsymbol{V}^{\mathtt{priv}}\right); (\boldsymbol{U}^*, \boldsymbol{V}^*)\right) \leq \left\|\Pi^{\mathtt{priv}} - \Pi^*\right\|_2^2 \cdot \mu^2 k \lambda_k + \left(\frac{k}{m}\right) \sigma_{\mathrm{F}}^2 \tag{24}
$$

Plugging in the bounds from Lemma 4.4 (and instantiating via Lemma 4.6) completes the proof. $\square$

## B.5 Proof of Lemma 4.4

*Proof.* Consider the $t$-th iteration of Algorithm 1. We first simplify the notation, i.e., let $\boldsymbol{U} = \boldsymbol{U}^{(t)}$ and $\boldsymbol{U}^+ = \boldsymbol{U}^{(t+1)}$, $\boldsymbol{v}_j = \boldsymbol{v}_j^{(t)}$.

Now, the clipping parameters are set large enough so that under the data generation assumptions (Assumption 4.1), there is no "clipping". So the updates in the Algorithm 1 and Algorithm 2 reduce to:

$$\boldsymbol{v}_j = \left( \frac{2}{m} \sum_{i \in [m/2]} \boldsymbol{U}^\top \mathbf{x}_{ij} \mathbf{x}_{ij}^\top \boldsymbol{U} \right)^{-1} \left( \frac{2}{m} \sum_{i \in [m/2]} y_{ij} \cdot \boldsymbol{U}^\top \mathbf{x}_{ij} \right),$$

$$\boldsymbol{H}^{(j)} = \frac{2}{m} \sum_{i \in [m/2+1,m]} \mathbf{x}_{ij} \mathbf{x}_{ij}^\top,$$

$$\boldsymbol{r}^{(t)} = \sum_{j \in \mathcal{S}_t} \left( \frac{2}{m} \sum_{i \in [m/2+1,m]} \mathbf{x}_{ij} \boldsymbol{z}_{ij} \right) \boldsymbol{v}_j^\top + \boldsymbol{g}^{(t)},$$

$$\widehat{\boldsymbol{U}} = \widetilde{\mathcal{A}}^{-1} \left( \sum_{j \in \mathcal{S}_t} \boldsymbol{H}^{(j)} \boldsymbol{U}^* \boldsymbol{v}_j^* \boldsymbol{v}_j^\top + \boldsymbol{r}^{(t)} \right),$$

$$\boldsymbol{U}^+ = \widehat{\boldsymbol{U}} \boldsymbol{R}^{-1}, \tag{25}$$

where $\boldsymbol{U}^+$ and $\boldsymbol{R}$ are obtained by QR decomposition of $\widehat{\boldsymbol{U}}$. Also, $\boldsymbol{g}^{(t)} \sim \eta \cdot \zeta \Delta_{(\varepsilon,\delta)} \cdot \mathcal{N}(0,1)^{dk}$, and $\widetilde{\mathcal{A}} : \mathbb{R}^{d \times k} \to \mathbb{R}^{d \times k}$ is defined as:

$$\widetilde{\mathcal{A}}(\boldsymbol{U}) = \mathcal{A}(\boldsymbol{U}) + \mathcal{G}(\boldsymbol{U}) \text{ with}$$

$$\mathcal{A}(\boldsymbol{U}) = \frac{2}{m} \sum_{i \in [m/2+1,m]} \boldsymbol{H}^{(j)} \boldsymbol{U} \boldsymbol{v}_j \boldsymbol{v}_j^\top, \text{ and } \mathcal{G}(\boldsymbol{U}) = \sum_{ab} \langle \boldsymbol{G}_{ab}, \boldsymbol{U} \rangle \mathbf{e}_a \mathbf{e}_b^\top,$$

where $\mathbf{e}_a$ is the $a$-th standard canonical basis vector, and for $\overrightarrow{\boldsymbol{G}_{ab}}$ being the vectorized version of $\boldsymbol{G}_{ab}$, $\bar{\boldsymbol{G}} = [\overrightarrow{\boldsymbol{G}_{11}}; \overrightarrow{\boldsymbol{G}_{12}}; \ldots; \overrightarrow{\boldsymbol{G}_{ab}}; \ldots \overrightarrow{\boldsymbol{G}_{dk}}] \sim \eta \zeta \Delta_{(\varepsilon,\delta)} \cdot \mathcal{N}_{\mathsf{sym}}(0,1)^{dk \times dk}$. Note that $\mathcal{A}$ and $\mathcal{G}$, and consequently $\widetilde{\mathcal{A}}$, are self-adjoint operator i.e. $\langle \widetilde{\mathcal{A}}(\boldsymbol{U}), \bar{\boldsymbol{U}} \rangle = \langle \boldsymbol{U}, \widetilde{\mathcal{A}}(\bar{\boldsymbol{U}}) \rangle$ for all $\boldsymbol{U}, \bar{\boldsymbol{U}}$. Furthermore, let $\mathcal{W}(\boldsymbol{U}) = \boldsymbol{U} \sum_j \boldsymbol{v}_j \boldsymbol{v}_j^\top$.

Note that the update for $\boldsymbol{v}_j$ is same as the update in the non-private Alternating Minimization algorithm (similar to Algorithm 1 of [46]). Now, let $\boldsymbol{Q} = (\boldsymbol{U}^*)^\top \boldsymbol{U}$, and $\Delta \in \mathbb{R}^{d \times k}$ be such that $\Delta_j = \boldsymbol{v}_j - \boldsymbol{Q}^{-1} \boldsymbol{v}_j^*$. Using Lemma B.4, we get:

$$\|\boldsymbol{v}_j\|_2 \leq \widetilde{O} \left( \frac{\mu^2 k}{n} \lambda_k^t \right), \quad \lambda_k \leq 2\lambda_k^t,$$

$$\max_j \|\Delta_j\|_2 \leq \widetilde{O} \left( \|(\mathbb{I} - \boldsymbol{U}^*(\boldsymbol{U}^*)^\top)\boldsymbol{U}\|_2 \cdot \mu \sqrt{k\lambda_k} \right) + \sigma_{\mathbb{F}} \sqrt{\frac{k \log n}{m}}, \tag{26}$$

where $\lambda_i^t$ is the $i$-th eigenvalue of $\frac{1}{n} \sum_j \boldsymbol{v}_j \boldsymbol{v}_j^\top$.

Now, using standard calculations, we get:

$$\widehat{\boldsymbol{U}} - \boldsymbol{U}^* \boldsymbol{Q} \tag{27}$$

$$= \widetilde{\mathcal{A}}^{-1} \left( \sum_j \boldsymbol{H}^{(j)} \boldsymbol{U}^* \boldsymbol{Q} (\boldsymbol{Q}^{-1} \boldsymbol{v}_j^* - \boldsymbol{v}_j) \boldsymbol{v}_j^\top + \sum_{ij} \boldsymbol{z}_{ij} \mathbf{x}_{ij} \boldsymbol{v}_j^\top + \boldsymbol{g}^{(t)} - \mathcal{G}(\boldsymbol{U}^* \boldsymbol{Q}) \right)$$

$$= \mathcal{W}^{-\frac{1}{2}} \left( \mathcal{W}^{\frac{1}{2}} \widetilde{\mathcal{A}}^{-1} \mathcal{W}^{\frac{1}{2}} \right) \mathcal{W}^{-\frac{1}{2}} \left( \sum_j \boldsymbol{H}^{(j)} \boldsymbol{U}^* \boldsymbol{Q} (\boldsymbol{Q}^{-1} \boldsymbol{v}_j^* - \boldsymbol{v}_j) \boldsymbol{v}_j^\top + \sum_{ij} \boldsymbol{z}_{ij} \mathbf{x}_{ij} \boldsymbol{v}_j^\top + \boldsymbol{g}^{(t)} - \mathcal{G}(\boldsymbol{U}^* \boldsymbol{Q}) \right)$$

$$= \boldsymbol{U}^* \boldsymbol{Q} \sum_j (\boldsymbol{Q}^{-1} \boldsymbol{v}_j^* - \boldsymbol{v}_j) \boldsymbol{v}_j^\top \left( \sum_j \boldsymbol{v}_j \boldsymbol{v}_j^\top \right)^{-1} + \boldsymbol{F} + \widetilde{\boldsymbol{F}}, \tag{28}$$

where for $\mathcal{E} = \mathcal{W}^{\frac{1}{2}}\widetilde{\mathcal{A}}^{-1}\mathcal{W}^{\frac{1}{2}} - I$,

$$\boldsymbol{F} = \mathcal{W}^{-\frac{1}{2}}\mathcal{E}\mathcal{W}^{-\frac{1}{2}}\left(\boldsymbol{U}^*\boldsymbol{Q}(\boldsymbol{Q}^{-1}\boldsymbol{v}_j^* - \boldsymbol{v}_j)\boldsymbol{v}_j^\top\right)$$

$$+ \mathcal{W}^{-\frac{1}{2}}\left(\mathbb{I} + \mathcal{E}\right)\mathcal{W}^{-\frac{1}{2}}\left(\sum_j (\boldsymbol{H}^{(j)} - I)\boldsymbol{U}^*\boldsymbol{Q}(\boldsymbol{Q}^{-1}\boldsymbol{v}_j^* - \boldsymbol{v}_j)\boldsymbol{v}_j^\top + \sum_{ij}\boldsymbol{z}_{ij}\mathbf{x}_{ij}\boldsymbol{v}_j^\top\right),$$

$$\widetilde{\boldsymbol{F}} = \mathcal{W}^{-\frac{1}{2}}\left(\mathbb{I} + \mathcal{E}\right)\mathcal{W}^{-\frac{1}{2}}\left(\boldsymbol{g}^{(t)} - \mathcal{G}(\boldsymbol{U}^*\boldsymbol{Q})\right).$$

Using Lemma B.3 and the assumption on $n$, $\Delta_{(\varepsilon,\delta)}$, we get:

$$\|\mathcal{E}\|_F \leq \frac{1}{32}. \tag{29}$$

Furthermore, using Lemma B.6, setting $\kappa = \lambda_1/\lambda_k$, we get w.p. $\geq 1 - 1/n^{100}$,

$$\|\boldsymbol{F}\|_F \leq \widetilde{O}\left(\mu\log n \cdot \sqrt{\frac{\kappa dk^2 T}{mn}}\|(\mathbb{I} - \boldsymbol{U}^*(\boldsymbol{U}^*)^\top)\boldsymbol{U}\|_F\right) + \sqrt{\frac{\mu^2 dk T\log n}{mn}}\cdot\frac{\sigma_{\mathrm{F}}}{\sqrt{\lambda_k}}. \tag{30}$$

Finally, using Lemma B.7, we get w.p. $\geq 1 - 1/n^{100}$,

$$\left\|\widetilde{\boldsymbol{F}}\right\|_F \leq \widetilde{O}\left(\frac{(\sqrt{k}\eta^2 + \eta\zeta)\Delta_{(\varepsilon,\delta)}\sqrt{dk}}{n\lambda_k}\right). \tag{31}$$

That is, by setting $n = \widetilde{\Omega}\left(\frac{\lambda_1}{\lambda_k}\cdot\mu^2 dk + \Delta_{(\varepsilon,\delta)}\cdot\left(\mathrm{NSR}^2 + \mu^2 k\right)d^{3/2}\right)$ and $m = \widetilde{\Omega}\left((1 + \mathrm{NSR})\cdot k + k^2\right)$ (as per Assumption 4.1), we get:

$$\|\boldsymbol{F}\|_F \leq \frac{1}{64}, \left\|\widetilde{\boldsymbol{F}}\right\|_F \leq \frac{1}{64}.$$

Similarly, using $n$ and $m$ as specified in Assumption 4.1 and Lemma B.6, for $\boldsymbol{M} = \boldsymbol{U}^*\boldsymbol{Q}\sum_j(\boldsymbol{Q}^{-1}\boldsymbol{v}_j^* - \boldsymbol{v}_j)\boldsymbol{v}_j^\top\left(\sum_j\boldsymbol{v}_j\boldsymbol{v}_j^\top\right)^{-1}$, we get

$$\|\boldsymbol{M}\|_F \leq \frac{1}{64}.$$

Finally, due to the initialization condition, $\sigma_{min}(\boldsymbol{Q}) \geq 1/2$. Thus, using standard calculations (for example, see Lemma A.3 in [46]), we get:

$$\|\boldsymbol{R}^{-1}\| \leq 4,$$

where $\widehat{\boldsymbol{U}} = \boldsymbol{U}^+\boldsymbol{R}$.

Note that $\boldsymbol{U}^*\boldsymbol{Q}\sum_j(\boldsymbol{Q}^{-1}\boldsymbol{v}_j^* - \boldsymbol{v}_j)\boldsymbol{v}_j^\top\left(\sum_j\boldsymbol{v}_j\boldsymbol{v}_j^\top\right)^{-1}$ lies along $\boldsymbol{U}^*$, so does not contribute to the error $\left\|(I - \boldsymbol{U}^*(\boldsymbol{U}^*)^\top)\boldsymbol{U}^+\right\|_F$. Hence,

$$\left\|(\mathbb{I} - \boldsymbol{U}^*(\boldsymbol{U}^*)^\top)\boldsymbol{U}^+\right\|_F \leq \left\|\boldsymbol{F} + \widetilde{\boldsymbol{F}}\right\|_F\left\|\boldsymbol{R}^{-1}\right\|_F \leq 4\left\|\boldsymbol{F} + \widetilde{\boldsymbol{F}}\right\|_F$$

$$\leq 4\widetilde{O}\left(\mu\log n \cdot \sqrt{\frac{\kappa dk^2 T}{mn}}\|(\mathbb{I} - \boldsymbol{U}^*(\boldsymbol{U}^*)^\top)\boldsymbol{U}\|_F + \sqrt{\frac{\mu^2 dk T\log n}{mn}}\cdot\frac{\sigma_{\mathrm{F}}}{\sqrt{\lambda_k}} + \frac{(\sqrt{k}\eta^2 + \eta\zeta)\Delta_{(\varepsilon,\delta)}\sqrt{dk}}{n\lambda_k}\right),$$

$$\leq \frac{1}{4}\left\|(\mathbb{I} - \boldsymbol{U}^*(\boldsymbol{U}^*)^\top)\boldsymbol{U}\right\|_F + \widetilde{O}\left(\sqrt{\frac{\mu^2 dk T\log n}{mn}}\cdot\frac{\sigma_{\mathrm{F}}}{\sqrt{\lambda_k}} + \frac{(\sqrt{k}\eta^2 + \eta\zeta)\Delta_{(\varepsilon,\delta)}\sqrt{dk}}{n\lambda_k}\right). \tag{32}$$

The result now follows by applying the above bound for all $t$ and by using: $\eta = \widetilde{O}(\mu\sqrt{\lambda_k dk})$, $\zeta = \widetilde{O}\left(\sigma_{\mathrm{F}} + \mu\sqrt{k\lambda_k}\right)$, i.e., $\sqrt{k}\eta^2 + \eta\zeta = \lambda_k\widetilde{O}((\mathrm{NSR} + \mu\sqrt{dk^2})\mu\sqrt{dk})$. $\qquad\square$

**Lemma B.3.** *Consider the setting of Lemma 4.4 and the notation introduced in the proof above. Let $\mathcal{E} = \mathcal{W}^{\frac{1}{2}}\widetilde{\mathcal{A}}^{-1}\mathcal{W}^{\frac{1}{2}} - I$. Then, w.p. $\geq 1 - 1/n^{100}$: $\|\mathcal{E}\|_F \leq \frac{1}{32}$.*

*Proof.* Using Lemma B.5 and (26), we get: $\|\mathcal{W}^{-\frac{1}{2}}\mathcal{A}\mathcal{W}^{-\frac{1}{2}} - \mathcal{I}\|_F \le 1/32$, where $\mathcal{I}(\boldsymbol{U}) = \boldsymbol{U}$. Furthermore, $\|\mathcal{W}^{-\frac{1}{2}}\mathcal{G}\mathcal{W}^{-\frac{1}{2}}\|_F \le 8\Delta_{(\varepsilon,\delta)}\sqrt{k}\eta^2\sqrt{\frac{dk}{n\lambda_k}}$ by using the bound on $\lambda_k^t$ given in (26). The result now follows by combining the above two given bounds. $\qquad\square$

**Lemma B.4** (Restatement of Lemma A.1 of [46]). *Consider the setting of Lemma 4.4 and the notation introduced in the proof above. Then, if $\|(I - \boldsymbol{U}^*(\boldsymbol{U}^*)^\top)\boldsymbol{U}\| \le \widetilde{O}(\frac{\lambda_k}{\lambda_1})$ and if $m \ge \widetilde{\Omega}\left((1 + \text{NSR}) \cdot k + k^2\right)$, we have w.p. $\ge 1 - 1/n^{101}$:*

$$\|\boldsymbol{v}_j\|_2 \le \widetilde{O}\left(\frac{\mu^2 k}{n}\lambda_k^t\right), \quad \lambda_k \le 2\lambda_k^t,$$

$$\max_j \|\Delta_j\|_2 \le \widetilde{O}\left(\|(I - \boldsymbol{U}^*(\boldsymbol{U}^*)^\top)\boldsymbol{U})\|_2 \cdot \mu\sqrt{k\lambda_k}\right) + \sigma_F\sqrt{\frac{k\log n}{m}}.$$

**Lemma B.5** (Restatement of Lemma A.7 of [46]). *Consider the setting of Lemma 4.4 and the notation introduced in the proof above. Let $mn \ge \widetilde{O}(\mu^2 dk^2)$, then w.p. $\ge 1 - 1/n^{100}$:*

$$\|\mathcal{E}\|_F \le \widetilde{O}\left(\sqrt{\frac{\mu^2 dk^2}{mn}}\right).$$

**Lemma B.6** (Restatement of Lemma A.2 of [46]). *Consider the setting of Lemma 4.4 and the notation introduced in the proof above. Then, if $mn \ge \widetilde{O}(\mu^2 dk^2)$, we have (w.p. $\ge 1 - 1/n^{80}$):*

$$\left\|\boldsymbol{U}^*\boldsymbol{Q}\sum_j(\boldsymbol{Q}^{-1}\boldsymbol{v}_j^* - \boldsymbol{v}_j)\boldsymbol{v}_j^\top\left(\sum_j \boldsymbol{v}_j\boldsymbol{v}_j^\top\right)^{-1}\right\|_F \le \widetilde{O}\left(\sqrt{\kappa}\|(I - \boldsymbol{U}^*(\boldsymbol{U}^*)^\top)\boldsymbol{U}\|_F + \frac{\sigma_F}{\sqrt{\lambda_k}}\cdot\sqrt{\frac{k}{m}}\right),$$

$$\|\boldsymbol{F}\|_F \le \widetilde{O}\left(\mu\log n \cdot \sqrt{\frac{\kappa dk^2 T}{mn}}\|(I - \boldsymbol{U}^*(\boldsymbol{U}^*)^\top)\boldsymbol{U}\|_F\right) + \sqrt{\frac{\mu^2 dkT\log n}{mn}}\cdot\frac{\sigma_F}{\sqrt{\lambda_k}}.$$

**Lemma B.7.** *Consider the setting of Lemma 4.4 and the notation introduced in the proof above. Let $\|\mathcal{E}\| \le 1/2$. Then, w.p. $\ge 1 - 1/n^{100}$:*

$$\left\|\widetilde{\boldsymbol{F}}\right\|_F \le \widetilde{O}\left(\frac{(\sqrt{k}\eta^2 + \eta\zeta)\Delta_{(\varepsilon,\delta)}\sqrt{dk}}{n\lambda_k}\right).$$

*Proof.* Note that,

$$\left\|\widetilde{\boldsymbol{F}}\right\|_F \le \|\mathcal{W}^{-\frac{1}{2}}(I + \mathcal{E})\mathcal{W}^{-\frac{1}{2}}\|_2 \cdot \|\boldsymbol{g}^{(t)} - \mathcal{G}(\boldsymbol{U}^*\boldsymbol{Q})\|_2 \le \frac{2}{n\lambda_k}(\|\boldsymbol{g}^{(t)}\|_2 + \|\mathcal{G}(\boldsymbol{U}^*\boldsymbol{Q})\|_F)$$

$$\le \frac{2}{n\lambda_k}(\|\boldsymbol{g}^{(t)}\|_2 + \sqrt{k}\|\boldsymbol{G}\|_2). \tag{33}$$

The lemma now follows by using the fact that: $\|\boldsymbol{g}^{(t)}\|_2 \le \widetilde{O}(\eta\zeta\sqrt{dk})$ and $\|\boldsymbol{G}\|_2 \le \widetilde{O}(\eta^2\sqrt{dk})$ with probability $1 - 1/n^{100}$. $\qquad\square$

## C   Missing Proofs from Section 5

*Proof of Theorem 5.1.* We are going to proof that the sampling step in Algorithm 4 guarantees $\varepsilon$-DP. Let $S_0(D) = \sum_{j\in[n]}\frac{2}{m}\sum_{i\in[m/2]}\ell\left(\langle\text{clip}\left(\boldsymbol{U}_0^\top\mathbf{x}_{ij}; L_f\right), \boldsymbol{v}_0; y_{ij}\rangle\right)$, where $\boldsymbol{U}_0$ is fixed rank-$k$ matrix with orthonormal columns in $\mathbb{R}^{d\times k}$, and $\boldsymbol{v}_0 \in \mathbb{R}^k, \|\boldsymbol{v}_0\|_2 \le C$ is a fixed vector. The sampling step in Algorithm 4 is identical to the following

$$\mathbf{Pr}[\boldsymbol{U}^{\text{priv}} = \boldsymbol{U}] \propto \exp\left(-\frac{\varepsilon}{8L_f C\xi}\cdot(\text{score}(\boldsymbol{U}) - S_0(D))\right). \tag{34}$$

Let $\mathcal{L}(\boldsymbol{U}; D) = \text{score}(\boldsymbol{U}) - S_0(D)$. Consider any neighboring data sets $D$ and $D'$ such that user $j$ in $D$ is replace by user $j'$ in $D'$. We now bound the sensitivity $\mathcal{L}(\boldsymbol{U}; D) - \mathcal{L}(\boldsymbol{U}; D')$. We have

$$\mathcal{L}(\boldsymbol{U}; D) - \mathcal{L}(\boldsymbol{U}; D')$$

$$= \left[ \min_{\|\boldsymbol{v}_j\|_2 \le C} \frac{2}{m} \sum_i \ell\left( \langle \text{clip}\left(\boldsymbol{U}^\top \mathbf{x}_{ij}; L_f\right), \boldsymbol{v}_j \rangle; y_{ij} \right) - \frac{2}{m} \sum_i \ell\left( \langle \text{clip}\left(\boldsymbol{U}_0^\top \mathbf{x}_{ij}; L_f\right), \boldsymbol{v}_0 \rangle; y_{ij} \right) \right]$$

$$- \left[ \min_{\|\boldsymbol{v}_{j'}\|_2 \le C} \frac{2}{m} \sum_i \ell\left( \langle \text{clip}\left(\boldsymbol{U}^\top \mathbf{x}_{ij'}; L_f\right), \boldsymbol{v}_{j'} \rangle; y_{ij'} \right) - \frac{2}{m} \sum_i \ell\left( \langle \text{clip}\left(\boldsymbol{U}_0^\top \mathbf{x}_{ij'}; L_f\right), \boldsymbol{v}_0 \rangle; y_{ij'} \right) \right]$$
(35)

Consider the first term. Let $\boldsymbol{v}_j^*$ be the minimizer of the first term. We have

$$\frac{2}{m} \sum_i \left( \ell\left( \langle \text{clip}\left(\boldsymbol{U}^\top \mathbf{x}_{ij}; L_f\right), \boldsymbol{v}_j^* \rangle; y_{ij} \right) - \ell(\langle \text{clip}\left(\boldsymbol{U}_0^\top \mathbf{x}_{ij}; L_f\right), \boldsymbol{v}_0 \rangle; y_{ij}) \right)$$

$$\le \frac{2}{m} \sum_i \xi \left| \langle \text{clip}\left(\boldsymbol{U}^\top \mathbf{x}_{ij}; L_f\right), \boldsymbol{v}_j^* \rangle - \langle \text{clip}\left(\boldsymbol{U}_0^\top \mathbf{x}_{ij}; L_f\right), \boldsymbol{v}_0 \rangle \right|$$

$$\le \frac{2}{m} \sum_i \xi \left( \left\| \text{clip}\left(\boldsymbol{U}^\top \mathbf{x}_{ij}; L_f\right) \right\|_2 \left\| \boldsymbol{v}_j^* \right\|_2 + \left\| \text{clip}\left(\boldsymbol{U}_0^\top \mathbf{x}_{ij}; L_f\right) \right\|_2 \left\| \boldsymbol{v}_0 \right\|_2 \right)$$

$$\le 2 \xi L_f C,$$

where the first inequality follows because $\ell$ is $\xi$-Lipschitz in the first parameter, and the last inequality follows from the bound on the norm of $\boldsymbol{v}$. Similar can be shown for the second term of (35). Therefore, the sensitivity of the score function, i.e. (35), is upper bounded by $4\xi L_f C$.

The rest of the proof follows from standard exponential mechanism argument [35]. $\qquad \square$

*Proof of Theorem 5.2.* First, to bound the size of the net $\mathcal{N}^\phi$ we use classic covering number bound from [6, Lemma 3.1]. We have $\left| \mathcal{N}^\phi \right| = O\left( \left( \frac{9\sqrt{k}}{\phi} \right)^{(2d+1)\cdot k} \right)$, since $\| \cdot \|_F$ of the matrices, over which the net is built, is $\sqrt{k}$. Let $\boldsymbol{U}^* = \arg\min_{\boldsymbol{U} \in \mathcal{K}} \text{score}(\boldsymbol{U})$.

First, we show that $\text{score}\left(\widetilde{\boldsymbol{U}}\right) - \text{score}\left(\boldsymbol{U}^*\right)$ is small for any $\widetilde{\boldsymbol{U}} \in \mathcal{N}^\phi$. For any $\widetilde{\boldsymbol{U}}$, we have,

$$\text{score}\left(\widetilde{\boldsymbol{U}}\right) \le \text{score}\left(\boldsymbol{U}^*\right) + \xi C \sum_{j \in [n]} \frac{2}{m} \sum_{i \in [m/2]} \left\| \text{clip}\left( \widetilde{\boldsymbol{U}}^\top \mathbf{x}_{ij}; L_f \right) - \text{clip}\left( (\boldsymbol{U}^*)^\top \mathbf{x}_{ij}; L_f \right) \right\|_2$$

$$= \text{score}\left(\boldsymbol{U}^*\right) + \xi C \sum_{j \in [n]} \frac{2}{m} \sum_{i \in [m/2]} \left\| \left( \widetilde{\boldsymbol{U}} - \boldsymbol{U}^* \right)^\top \mathbf{x}_{ij} \right\|_2, \qquad (36)$$

with probability $\ge 1 - 1/n^{10}$. The first step follows from the Lipschitzness of $\ell$ and $\|\boldsymbol{v}\|_2 \le C$, and the second step follows because the choice of $L_f$ will not introduce any effect due to clipping w.p. at least $1 - \frac{1}{n^{10}}$. We will condition the rest of the analysis on this.

Let $\boldsymbol{M} = \widetilde{\boldsymbol{U}} - \boldsymbol{U}^*$ with columns $[\boldsymbol{m}_a : a \in [k]]$. By the definition of the net, we have $\sum_{a=1}^{k} \|\boldsymbol{m}_a\|_2^2 \le \phi^2$. Since the feature vectors are drawn i.i.d. from $\mathcal{N}(0,1)^d$, we have $\langle \boldsymbol{m}_a, \mathbf{x}_{ij} \rangle \sim \mathcal{N}\left( 0, \|\boldsymbol{m}_a\|_2^2 \right)$. Therefore, by standard Gaussian concentration and union bound, we have w.p. at least $1 - \frac{1}{n^{10}}$, $\forall i \in [m/2], j \in [n], a \in [k], |\langle \boldsymbol{m}_a, \mathbf{x}_{ij} \rangle| \le \|\boldsymbol{m}_a\|_2 \cdot \text{polylog}(n)$. Therefore, $\left\| \boldsymbol{M}^\top \mathbf{x}_{ij} \right\|_2 \le \phi \cdot \text{polylog}(n)$. Substituting back to (36), we have

$$\text{score}\left(\widetilde{\boldsymbol{U}}\right) \le \text{score}\left(\boldsymbol{U}^*\right) + \xi C n \phi \cdot \text{polylog}(n). \qquad (37)$$

Second, we aim to show that $\boldsymbol{U}^{\mathrm{priv}}$ and $\widetilde{\boldsymbol{U}}$ are close. For any $\gamma$, we have

$$
\mathbf{Pr}\left[\mathsf{score}\left(\boldsymbol{U}^{\mathrm{priv}}\right) - \mathsf{score}\left(\widetilde{\boldsymbol{U}}\right) \geq \gamma\right] \leq |\mathcal{N}^{\phi}| \cdot \frac{\exp\left(-\frac{\varepsilon}{8\xi L_f C} \cdot \left(\mathsf{score}\left(\widetilde{\boldsymbol{U}}\right) + \gamma\right)\right)}{\exp\left(-\frac{\varepsilon}{8\xi L_f C} \cdot \mathsf{score}\left(\widetilde{\boldsymbol{U}}\right)\right)}
$$

$$
= |\mathcal{N}^{\phi}| \cdot \exp\left(-\frac{\varepsilon\gamma}{8\xi L_f C}\right). \tag{38}
$$

Setting $\gamma$ appropriately, we have w.p. at least $1 - \beta$,

$$
\mathsf{score}\left(\boldsymbol{U}^{\mathrm{priv}}\right) - \mathsf{score}\left(\widetilde{\boldsymbol{U}}\right) \leq \frac{8\xi CL_f \log\left(|\mathcal{N}^{\phi}|/\beta\right)}{\varepsilon} = O\left(\frac{\xi CL_f dk}{\varepsilon}\log\left(\frac{k}{\phi\beta}\right)\right). \tag{39}
$$

Now we show a bound on the excess empirical risk. Combining (37) and (39), we have

$$
\mathsf{score}\left(\boldsymbol{U}^{\mathrm{priv}}\right) \leq \mathsf{score}\left(\boldsymbol{U}^{*}\right) + O\left(\frac{\xi CL_f dk}{\varepsilon}\log\left(\frac{k}{\phi\beta}\right) + \xi Cn\phi \cdot \mathrm{polylog}\left(n\right)\right).
$$

Let $\mathcal{L}_{\mathrm{ERM}}(\boldsymbol{U}, \boldsymbol{V}) = \frac{2}{mn}\sum_{i\in[m/2],j\in[n]}\ell\left(\langle\boldsymbol{U}^{\top}\mathbf{x}_{ij}, \boldsymbol{v}_j\rangle; y_{ij}\right)$, and $\widehat{\boldsymbol{V}} = \min_{\boldsymbol{V}}\mathcal{L}_{\mathrm{ERM}}(\boldsymbol{U}^{\mathrm{priv}}, \boldsymbol{V})$, i.e., the minimizer for $\mathsf{score}\left(\boldsymbol{U}^{\mathrm{priv}}\right)$. The above inequality directly transfers to

$$
\mathcal{L}_{\mathrm{ERM}}(\boldsymbol{U}^{\mathrm{priv}}, \widehat{\boldsymbol{V}}) \leq \mathcal{L}_{\mathrm{ERM}}(\boldsymbol{U}^{*}, \boldsymbol{V}^{*}) + O\left(\frac{\xi CL_f \cdot dk}{\varepsilon n}\log\left(\frac{k}{\phi\beta}\right) + \xi C\phi \cdot \mathrm{polylog}\left(n\right)\right) \tag{40}
$$

Setting $\phi = \frac{1}{\varepsilon n}$ and plugging in $L_f = O(\sqrt{d}\log(nm))$, the above inequality becomes,

$$
\mathcal{L}_{\mathrm{ERM}}(\boldsymbol{U}^{\mathrm{priv}}, \widehat{\boldsymbol{V}}) \leq \mathcal{L}_{\mathrm{ERM}}(\boldsymbol{U}^{*}, \boldsymbol{V}^{*}) + O\left(\frac{\xi C\sqrt{k^2 d^3}}{\varepsilon n}\right) \cdot \mathrm{polylog}\left(n\right). \tag{41}
$$

Finally, to complete the proof, we need to translate the excess empirical risk bound into excess population risk bound. Recall the following definition of population risk.

$$
\mathcal{L}_{\mathrm{Pop}}(\boldsymbol{U}; \boldsymbol{V}) = \mathbb{E}_{(i,j)\sim_u[m/2]\times[n],(\mathbf{x}_{ij},y_{ij})\sim\tau}\left[\ell\left(\langle\boldsymbol{U}^{\top}\mathbf{x}_{ij}, \boldsymbol{v}_j\rangle; y_{ij}\right)\right] \tag{42}
$$

We have the following.

$$
\mathcal{L}_{\mathrm{Pop}}(\boldsymbol{U}^{\mathrm{priv}}; \boldsymbol{V}^{\mathrm{priv}}) - \mathcal{L}_{\mathrm{Pop}}(\boldsymbol{U}^{*}, \boldsymbol{V}^{*})
$$
$$
= \left(\mathcal{L}_{\mathrm{Pop}}(\boldsymbol{U}^{\mathrm{priv}}; \boldsymbol{V}^{\mathrm{priv}}) - \mathcal{L}_{\mathrm{Pop}}(\boldsymbol{U}^{\mathrm{priv}}, \boldsymbol{V}^{*})\right) + \left(\mathcal{L}_{\mathrm{Pop}}(\boldsymbol{U}^{\mathrm{priv}}, \boldsymbol{V}^{*}) - \mathcal{L}_{\mathrm{Pop}}(\boldsymbol{U}^{*}, \boldsymbol{V}^{*})\right) \tag{43}
$$

We will bound the two terms separately. For the first term $\mathcal{L}_{\mathrm{Pop}}(\boldsymbol{U}^{\mathrm{priv}}, \boldsymbol{V}^{\mathrm{priv}}) - \mathcal{L}_{\mathrm{Pop}}(\boldsymbol{U}^{\mathrm{priv}}, \boldsymbol{V}^{*})$, notice that $\boldsymbol{U}^{\mathrm{priv}}$ and $\boldsymbol{V}^{\mathrm{priv}}$ are independent as they are trained on disjoint data. This implies $\forall i \in \{m/2+1, \cdots, m\}, j \in [n]$, w.p. at least $1 - \frac{1}{\min\{d,n\}^{10}}$, $\left\|\left(\boldsymbol{U}^{\mathrm{priv}}\right)^{\top}\mathbf{x}_{ij}\right\|_2 \leq \sqrt{k} \cdot \mathrm{polylog}\left(d, n\right)$. Since the loss functions have the form $\ell(\langle\left(\boldsymbol{U}^{\mathrm{priv}}\right)^{\top}\mathbf{x}, \boldsymbol{v}\rangle; y)$, by standard uniform convergence bound [2], we have the following.

$$
\mathcal{L}_{\mathrm{Pop}}(\boldsymbol{U}^{\mathrm{priv}}, \boldsymbol{V}^{\mathrm{priv}}) - \mathcal{L}_{\mathrm{Pop}}(\boldsymbol{U}^{\mathrm{priv}}, \boldsymbol{V}^{*}) = O\left(\xi C\sqrt{\frac{k}{m}}\right) \cdot \mathrm{polylog}\left(d, n\right) \tag{44}
$$

Then we bound the second term $\mathcal{L}_{\mathrm{Pop}}(\boldsymbol{U}^{\mathrm{priv}}, \boldsymbol{V}^{*}) - \mathcal{L}_{\mathrm{Pop}}(\boldsymbol{U}^{*}, \boldsymbol{V}^{*})$ in (43). We can write the inner product $\langle\boldsymbol{U}^{\top}\mathbf{x}, \boldsymbol{v}\rangle$ as $\langle\boldsymbol{U}, \mathbf{x}\boldsymbol{v}^{\top}\rangle$. Therefore, if we vectorize $\boldsymbol{U}$ by concatenating its the columns as $\overrightarrow{\boldsymbol{U}}$, and vectorize $\mathbf{x}\boldsymbol{v}^{\top}$ by concatenating its columns as $\overrightarrow{\boldsymbol{z}}$, the inner product equals to $\langle\boldsymbol{z}, \overrightarrow{\boldsymbol{U}}\rangle$. The loss function can be written as $\ell(\langle\boldsymbol{U}^{\top}\mathbf{x}, \boldsymbol{v}\rangle; y) = \ell\left(\langle\boldsymbol{z}, \overrightarrow{\boldsymbol{U}}\rangle; y\right)$. We define $\boldsymbol{z}_{ij}$ as the vectorized version of $\mathbf{x}_{ij}(\boldsymbol{v}_j^*)^{\top}$. With probability at least $1 - \frac{1}{\min\{d,n\}^{10}}$, $\forall i \in [m/2], j \in [n], \|\boldsymbol{z}_{ij}\|_2 \leq$

$C\sqrt{d} \cdot \text{polylog}\,(d, n)$. By standard uniform convergence bound [2] and the bound on the empirical Rademacher complexity below, we have

$$\mathcal{L}_{\text{Pop}}(\boldsymbol{U}^{\text{priv}}, \boldsymbol{V}^*) - \mathcal{L}_{\text{Pop}}(\boldsymbol{U}^*, \boldsymbol{V}^*)$$

$$\leq \mathcal{L}_{\text{ERM}}(\boldsymbol{U}^{\text{priv}}, \widehat{\boldsymbol{V}}) - \mathcal{L}_{\text{ERM}}(\boldsymbol{U}^*, \boldsymbol{V}^*) + O\left(\xi C \sqrt{\frac{d}{nm}}\right) \cdot \text{polylog}\,(d, n). \tag{45}$$

Combining (41), (45), (44) into (43) and translating the high-probability to expectation statement completes the proof.

**Bound on Rademacher complexity:** We aim to compute the Rademacher complexity of $\langle \boldsymbol{U}, \sum_{ij} \mathbf{x}_{ij} \boldsymbol{v}_j^\top \rangle = \sum_{ij} \langle \mathbf{x}_{ij}, \boldsymbol{U} \boldsymbol{v}_j \rangle$. We will follow [33, Theorem 11] with small modification in the Cauchy-Schwartz step.

Let $\theta$ be a vector of length $nd$ that is formed by concatenating $\boldsymbol{U} \boldsymbol{v}_j$ for all $j$. For any $i, j$, let $\widetilde{\mathbf{x}}_{ij}$ be a vector of length $dn$, such that the $j$-th "block" (of length $d$) is $\mathbf{x}_{ij}$ and the rest of the entries are $0$. So we can express $\langle \mathbf{x}_{ij}, \boldsymbol{U} \boldsymbol{v}_j \rangle$ as $\langle \widetilde{\mathbf{x}}_{ij}, \theta \rangle$. We have

$$\langle \widetilde{\mathbf{x}}_{ij}, \theta \rangle = \langle \mathbf{x}_{ij}, \boldsymbol{U} \boldsymbol{v}_j \rangle \leq \|\mathbf{x}_{ij}\|_2 \|\boldsymbol{U} \boldsymbol{v}_j\|_2 \leq C \|\mathbf{x}_{ij}\|_2,$$

where the last step follows because $\boldsymbol{U}$ is orthonormal and $\|\boldsymbol{v}_j\|_2 \leq C$. Also, because the data is drawn from a normal distribution, we have $\mathbb{E}\left[\|\widetilde{\mathbf{x}}_{ij}\|_2^2\right] = \mathbb{E}\left[\|\mathbf{x}_{ij}\|_2^2\right] = d$. The Rademacher complexity is $\frac{C\sqrt{d}}{\sqrt{mn}}$ following the same argument as [33, Theorem 11]. $\qquad\square$