# OpenReview forum: "Differentially Private Model Personalization"
_NeurIPS.cc/2021/Conference — NeurIPS 2021 Spotlight_

### Official Review · Reviewer_2osN · 2021-07-14

**Rating:** 7
**Confidence:** 2

**Summary:**


The paper presents an algorithm for differentially private (DP) learning of personalized models. A set of users with their own private data collaborate to privately learn a good embedding of the data followed by a linear regression based on this embedding.

The paper contribution is a theoretical analysis of the proposed algorithm with DP guarantees and utility bounds.

As far as I know there is no such bounds in the literature within the same setting. The author compare their result with the bounds in the non private case and with an upper bound obtained using the exponential mechanism which provides an information theoretic bound but cannot be efficiently implemented. In the later case, the bounds on the excess risk differs by O(1/\epsilon) and O(1/\sqrt(m)), m being the number of samples and \epsilon the privacy budget.


**Limitations And Societal Impact:**

Sect. 6.

**Main Review:**

The paper is clearly written, although the notations are heavy, the contribution very technical and a bit hard to follow. The paper includes a description of the setting and a less formal presentation of the contributions that are very welcome. However, two points could be improved. A first regret concerns the presentation of the algorithms. There is no informal explanation after Section 3 and even if lines are mostly easy to understand, some choices and notation are difficult to understand. A second point is that a discussion of Assumptions 4.1 is missing.

The generic approach is presented in Section 3 and  instantiated with linear regression in Section 4. The point that is unclear to me is how the data sets are split to estimate U and v. It is clear that the choice was to take different subsets for both estimations but this should be explicitly explained in the main text. Also, why the summation is over the first quarter of samples (i \in [m/4]) for v on line 5 and the second quarter for U on line 6, and not taking half of the data for each estimation is not explained (The same comment can also be made for Algorithm 2).

Thm 4.2 presents the main (utility) result, a bound on the excess risk of Algo 1. Algo 1 has \alpha,\rho  RDP parameters and the result depends on \epsilon and \delta. It should be clearer to have coherent parametrization for the two.

The bound depends on structural  parameters of the regression v. In an oversimplified case, when these parameters are close to  constants, Remark 1 and 2 give comparisons of Thm 4.2 with the non DP case. It is not clear in the general case how these parameters impact the bound.

Essentially, the proposed algorithm proceeds by alternating the optimization of the embedding and the optimization of the regression parameters in the embedded space and the privacy budget is split for the 2 subtasks. It is not discussed if the proposed split is optimal or not. Only the parallel composition is proposed and may not be suitable when the number of samples m is low, which is generally assumed in collaborative learning.

Additionally, a good starting point for the embedding is necessary to obtain Thm 4.2. This initialization step is proposed in Sect 4.2. Again a description of this algorithm is welcome. For instance why consecutive samples are used to estimate M^noisy and why U^init consists in its top eigenvectors.

Strengths
- Interesting problem
- Theoretically grounded
- Clear presentation

Weaknesses
- Very technical and proofs are difficult to follow
- Analysis conducted in the case of linear regression with independent features, which seems limited (and maybe limits the benefits of finding a common embedding)
- No experimentation


typos and comments
- l210 models
- I don't understand the notation ||v||_2 \leq R^k line 5 of Algorithm 1.
- l255: citation is missing
- Algorithm 3 should output U^init and not U^priv. Same in Lemma 4.6
- I've try to read the proof in the supplementary, but they are too technical for the time I had to make a review. For instance l503-514 are not sufficiently explained for me.


**Time Spent Reviewing:**

12

---

> ### Author Response · Authors · 2021-08-10
> **Response**
>
> Thank you for the thoughtful feedback. We will like to clarify the specific concerns raised in the review.
>
> 1. **No explanation in Section 3 and for Assumption 4.1**: Thanks for pointing this out. We will add more introductory text surrounding the Algorithm 1 which makes parsing the notation easier. With respect to Assumption 4.1, it is easiest to understand in the context where the personalization vectors ($v^*_j$) are drawn from i.i.d. Gaussian, and the feature vectors $x_\{ij\}$ are drawn from i.i.d. Gaussian. We did instantiate Assumption 4.1 in this setting in Remark 1 when we comment about the bound in Theorem 4.2. We will add an explicit remark specific to Assumption 4.1 in the Gaussian case mentioned above.
>
> 2. **Split of the data**: The data is split half and half for the estimation part of $U$ and for the estimation of $v$. In the estimation of $U$ in Algorithm 1, data is further divided into ½ and ½ between steps 5 and 6. (This split is mainly for technical reasons to decorrelate Steps 5 and 6, for the proof to go through.) We did not try to optimize for the splits to obtain our asymptotic guarantees. It is rightly mentioned in the review that the splits mentioned in the paper may not be optimal in practice. We will add a discussion on how one should go about choosing the splits, if they were to implement the algorithm in practice. In our recent experiments (see the comment to Reviewer AbPh), we did not do any split of the data, and used the same data for all the three parts. The resulting accuracy was fairly good.
>
> 3. **Translation between RDP and DP**: We used the RDP for privacy accounting purposes, so it was convenient for parametrizing the algorithm with it. In Theorem 4.2 will make the choice of RDP parameters for corresponding $(\epsilon,\delta)$-differential privacy explicit. Thank you for pointing this out.
>
> 4. **Dependence of the bound on structural parameters of $v$**: Our bound depends on incoherence parameter ($\mu$) which in some sense captures the structure of $v$. Our dependence on $\mu$ is similar to the dependence obtained by non-DP results in similar setting (e.g. [39]).
>
> 5. **More discussion about the initialization**: We will definitely add more discussion about the initialization step. In regards to the specific question of using consecutive samples for estimating M^noisy, we do so to make sure that we operate with only positive semidefinite matrices in the estimation. This makes the analysis a bit easier. A similar trick was also used in https://arxiv.org/abs/2002.08936 .
>
> 6. **Typos and other comments**: We will definitely fix all the typos and missing citations mentioned.  Specific to the comment about $||v||_2 \leq R^k$ line 5 of Algorithm 1, it should be $||v||_2 \in R^k$. We will also make the supplementary section more modular. Specific to the comment about Lines 503 to 514, the main insight there is that the Frobenius norm for matrices can be thought of as  $\ell_2$-norm for vectors, when the matrix is flattened. Hence, one can use standard Renyi differential privacy arguments for Gaussian distribution there. Lines 503 to 514 were capturing the associated calculations.

---

> > ### Comment · Reviewer_2osN · 2021-09-02
> > **Re: Response**
> >
> > Thanks for the clarifications

---

### Official Review · Reviewer_AbPh · 2021-07-15

**Rating:** 6
**Confidence:** 4

**Summary:**

This paper focuses on a setting, where there are multiple users each holds a training dataset. In this setting, users have to share information about their model or data to learn a meaningful representation and to solve their tasks better than they could without using the shared, low-dimensional representation. The paper proposes an approach for this problem that provides strong user-level differential privacy guarantees at the same time. For the privacy guaranteees, the method uses exponential mechanism and utility analysis of the algorithm is made by using that.

**Limitations And Societal Impact:**

As the authors already mentioned, the method has some limitations. The first question from my side is the reason of choosing exponential mechanism. There are some recent approaches such as "Privacy Amplification by Sampling and Renyi Differential Privacy" (Wang et al.) , "Privacy Amplification via Random Check-Ins" (Balle et al.), "Accuracy Gains from Privacy Amplification Through Sampling for Differential Privacy" (Hu et al.). These papers uses Gaussian mechanism and it has been proved that Gaussian mechanism is more efficient than exponential mechanism in iterative approaches. Could the authors please explain the reason of this choice?

The second one is the generalizability of the approach to deep networks. The proposed approach uses a two-layer neural network, where the first layer is shared across all users and the second layer is trained individually. I'm wondering how that method can be applied to different models. In the current version, it is a restrictive approach and it might not work/perform well on large or complex datasets as we encounter in real-world.

**Main Review:**

This paper focuses on an important problem which is protecting the user-privacy in a multi-task setting. The setting considers multiple users each holds a dataset and the authors introduces an algorithm for private model personalization. The contribution of the work and the problem definition are clear and the theoretical analysis of privacy and utility seems correct to me.

However, the main problem of this paper is lack of motivation. I couldn't understand how this method is different/better than any related published work in the literature. There are similar works proposed before, and the authors didn't discuss the similarities and the differences clearly. There are no experiments, so it is not possible to see how well this method performs on even toy dataset, so it is really difficult to foresee the success of the proposed approach. I will write in detail in the following section, but another concern is the limitations of the proposed approach.

I can suggest to the authors to explain the goals of the proposed method better, compare with the previous works and run some experiments on at least some toy datasets to show the practicality of the proposed method better. As a minor comment, I encounter some typos/grammatical mistakes that makes to follow the paper difficult, so it could be also improved in the future.

===================================================

In the rebuttal authors clarified my concern about comparison of this method with Gaussian mechanism. I read the paper again and read the other reviewers comments. In addition to that, they run some experiments and that further clarified my concerns. I'll update my score depending on these inputs.

**Time Spent Reviewing:**

8

---

> ### Author Response · Authors · 2021-08-10
> **Response**
>
> Thank you for the thoughtful feedback. We will like to clarify the specific concerns raised in the review.
>
> 1. **Motivation and comparison to prior work in the context of privacy**: In the related work section, we do a detailed comparison to all the work we are aware of in the context of private personalization [14,19,28,26,16,27,22,6]. The distinguishing features of our work, compared to prior work, are that we work with user-level privacy guarantees and provide the first privacy/utility trade-off for any non-trivial personalized learning task.
>
> 2. **Comment regarding exponential mechanism**: We believe that there has been an unfortunate confusion about our main method.  In the work, we propose and analyze the Priv-AltMin method (Algorithm 1, 2,3) which is indeed based on Gaussian additive noise.
> We analyze exponential mechanism to establish a “gold standard baseline” in terms of excess risk considered in this paper; note that the standard SGD+Gaussian noise style approaches that the reviewer mentions do not provide excess risk bounds for the highly non-convex loss functions considered. This is in line with the standard DP literature where exponential mechanism is often used to establish a “gold standard baseline” for the best achievable utility/privacy trade-offs under minimal assumptions [3].
> Our main result is Theorem 4.2 establishing error bounds for our Priv-AltMin method that uses Gaussian noise and we show that compared to exponential mechanism our method is significantly more efficient and has a better error bound, assuming certain data distribution properties.
>
> 3. **Generalizability of network architecture**: The two tier architecture is quite common for personalization and transfer learning literature (http://proceedings.mlr.press/v32/donahue14.html).  The top tier(s) is called the embedding layer, and the lower tier(s) is called the personalization layer. The formulation in the paper is generic enough to capture these architectures. Although we provide utility theorems that assume a specific generalized linear structure, the algorithms are applicable more generally.
>
> 4. **Experimentation**:
> To address the concern about empirical performance of our method, we conducted experiments for the two layer linear model setting provided in Section 4.
> We compared with the following strong baselines:
>     - `nonpriv`: None private training, i.e., the algorithm is run without clipping or additive noise.
>     - `own data`: each user’s learns their own model based on their own data. There is thus no privacy concern.
>     - `single`: the server learns a global $d$-dimensional model with user-level privacy.
>
>     We consider the setting where number of users $n = 50000$, number of examples per user $m = 10$, data dimensionality $d = 50$, and rank $k = 2$. Additionally, we consider data noise $\sigma_F = 0.01$, i.e, $y = \langle Uv, x\rangle + \mathcal{N}(0, \sigma_F^2)$. We report the MSE over two runs.
>
>     $\epsilon$|nonpriv|own data|single|our method
>     ---------|:-----:|:-----:|:-----:|:--------:
>     1.0 |0.00013|1.59986|2.08517|2.31232
>     2.0 |0.00013|1.59986|2.03179|1.09355
>     5.0 |0.00013|1.59986|2.01802|0.06929
>     10.0|0.00013|1.59986|2.01590|0.01794
>
>     Note that for $\epsilon=10$, which is commonly used in empirical evaluation, our method is able to get estimation error down to only 1e-2 while the "single" global privacy preserving method has error of 2.0 which is close to the error achieved by *any random parameter vector*. Similarly, if each user uses their own data, then also the error is close to the error of a random parameter vector.

---

> > ### Comment · Reviewer_AbPh · 2021-08-29
> > **Response #2**
> >
> > Many thanks for your response and clarifying the confusion. I rechecked the contribution carefully and realized that I misunderstood that point.

---

### Official Review · Reviewer_KGBe · 2021-07-16

**Rating:** 7
**Confidence:** 4

**Summary:**

This paper studies the problem of model personalization with user-level differential privacy in the setting where users don't have enough data to find a good solution on their own but can leverage information in similar learning problems shared by other users. The problem is viewed as learning a 2-layer neural network, where the first embedding is the shared structure and the second layer is trained individually. This paper provides two types of algorithms: inefficient algorithms based on the exponential mechanism that establish information-theoretic upper bounds on an achievable error and efficient algorithms based on alternating minimization, which starts from an initial embedding and then repeatedly uses a DP minimizer to minimize the error. For the specific case of linear regression with squared error loss, the paper shows the alternating minimization framework can converge to a near-optimal embedding.


**Limitations And Societal Impact:**

The authors provide several open problems that this work leads. This work has a positive social impact as it motivates practitioners and researchers to make their personalization ML systems differentially private.

**Main Review:**

I think this is a strong paper. It's a well-written paper. It studies an important yet remarkably understudied problem that has many applications. It gives two types of algorithms: exponential mechanism based model personalization and model personalization via private alternating minimization. The algorithm and the theoretical results are sound. Although the algorithm is fairly natural, the idea of formulating the problem as learning a 2-layer neural network is very novel, and the result is quite exciting--it shows we can accurately share a linear representation of the data while preserving user-level privacy, and the error bounds and sample complexity are nearly-optimal.

**Time Spent Reviewing:**

6

---

> ### Author Response · Authors · 2021-08-10
> **Response**
>
> Thank you for the thoughtful feedback and support for the paper. We will be happy to answer any questions you may have during the discussion phase.

---

### Official Review · Reviewer_XcxK · 2021-07-16

**Rating:** 6
**Confidence:** 3

**Summary:**

The authors deal with personalized model learning under the user-level joint differential privacy. In this problem, each user aims to build a user-specified model to make a prediction for the user's data. In the training process, the users utilize knowledge from the other users' data to improve their model's performance. In this setup, we want to prevent the leakage of the sensitive information in the user's data to the other user. The authors develop the differentially private algorithms for the model personalization task and reveal the upper bounds on the personalized model's accuracy of these algorithms.

**Limitations And Societal Impact:**

 The limitations and potential societal impacts are adequately discussed.

**Main Review:**

This paper is well-written and technically sound. My current decision is acceptance. However, the paper has several unclarities the authors should address.

The model personalization task and its privacy risk are well-motivated. The authors reveal a reasonable excess risk bound for their proposed method based on the exponential mechanism, which can be a significant technical contribution.

An unclarity comes from the upper bound shown in Theorem 4.2. The last term in this bound is a constant $\sigma_F^2$ and is dominated when $n$ and $m$ are large. Also, this result indicates that $\mathcal{A}_{\mathrm{Priv-AltMin}}$ can be inconsistent. Hence, I'm wondering why this bound is reasonable. Additionally, since the constant term dominates the bound as mentioned above, the error bound shown in Theorem 1.1 is misleading; the constant term should not be ignored.

Some notations are unclear. What is $f'$ in Lines 71-72? $\mathcal{C}$ in Line 77 is not defined.

It is better to compare the resultant upper bounds with the best bound for the method without the privacy guarantee. By such a comparison, we can see the tightness of the present bounds roughly.

**Time Spent Reviewing:**

7

---

> ### Author Response · Authors · 2021-08-10
> **Response**
>
> Thank you for the thoughtful feedback. We will like to clarify the specific concerns raised in the review.
> 1. **Comment regarding $\sigma_F$**: It is a typo. The second term in Line 247 should be just $(k/m)\sigma_F^2$ instead of $(k/m+1)\sigma_F^2$. Since we are looking at excess population risk (Eq 1), the true risk for the model $(U^*, v^*)$, which equals to $\sigma^2_F$ should be subtracted from the risk of the model $(U^{priv},V^{priv})$. Sorry for the confusion. We will fix it in the next version of the paper.
> 2. **Notation not defined**: $f’$ in Lines 71-72 represents the link function mapping one-dimensional input to an output, e.g., identity function or sigmoid function. $\mathcal{C}$ in Line 77 is a fixed set from which the personalization vectors $v_1,\ldots,v_n$ are drawn from. Thanks for pointing out these omissions! We will make these notations more clear.
> 3. **Comparison to best known non-private bounds**: We compare against non-private information theoretic bound in Line 263-264 (i.e. when $\epsilon\to\infty$), and show that it is only $O(k)$ worse. Furthermore, compared to the non-private method of  [39], our analysis has almost the same error bound except for the term dependent on privacy parameter $\epsilon$ and certain poly-log factors.

---

> > ### Comment · Reviewer_XcxK · 2021-09-02
> > **Re: Response**
> >
> > Thank you for the response and clarification. The response clarifies all of my concerns.

---

### Decision · Program_Chairs · 2021-09-27

**Decision:**

Accept (Spotlight)

**Comment:**

This paper analyzes a setting in which multiple users, each with their own dataset, wish to collaborate so as to learn better personalized model, while maintaining the privacy of their data. In the framework analyzed, all users share a common, low-dimensional embedding model, the output of which is used as the input for each personalized model. The paper presents two algorithms, based on two DP mechanisms, and proves that they are $(\epsilon, \delta)$-DP while also providing bounds on the excess population risk w.r.t. non-private learning -- i.e., the price paid for private learning.

This is a very interesting and timely topic at the intersection of federated learning, on-device personalization and differential privacy. The theoretical results are compelling and clearly presented. The writing is OK, although I noticed a few typos in the intro -- mostly missing articles. The reviews ended up being unanimously positive. One reviewer was originally against acceptance, but was later convinced to change their score by the authors' response.

I am recommending this paper for a spotlight because I believe that DP personalization is an important emerging topic that has not yet received much attention in the community. However, I recognize that the paper did not receive very high scores, so I am comfortable with the SAC/PCs downgrading it to a poster if need be.

Note that the submitted manuscript does not contain experimental results, but the authors provided some during discussion at the request of a reviewer. I encourage the authors to include these in the paper. (Experiments may strengthen the case for a spotlight -- or even a full oral.)

I also encourage the authors to give the paper a close read for grammar, syntax and clarity, as there were multiple complaints about the paper's clarity, and I myself think the writing could be polished.